# The transcriptional repressor complex FRS7-FRS12 regulates flowering time and growth in *Arabidopsis*

Andrés Ritter[1,2,*,†], Sabrina Iñigo[1,2,*], Patricia Fernández-Calvo[1,2,*], Ken S. Heyndrickx[1,2], Stijn Dhondt[1,2], Hua Shi[3], Liesbeth De Milde[1,2], Robin Vanden Bossche[1,2], Rebecca De Clercq[1,2], Dominique Eeckhout[1,2], Mily Ron[4], David E. Somers[3], Dirk Inzé[1,2], Kris Gevaert[5,6], Geert De Jaeger[1,2], Klaas Vandepoele[1,2], Laurens Pauwels[1,2] & Alain Goossens[1,2]

Most living organisms developed systems to efficiently time environmental changes. The plant-clock acts in coordination with external signals to generate output responses determining seasonal growth and flowering time. Here, we show that two *Arabidopsis thaliana* transcription factors, FAR1 RELATED SEQUENCE 7 (FRS7) and FRS12, act as negative regulators of these processes. These proteins accumulate particularly in short-day conditions and interact to form a complex. Loss-of-function of *FRS7* and *FRS12* results in early flowering plants with overly elongated hypocotyls mainly in short days. We demonstrate by molecular analysis that FRS7 and FRS12 affect these developmental processes in part by binding to the promoters and repressing the expression of *GIGANTEA* and *PHYTOCHROME INTERACTING FACTOR 4* as well as several of their downstream signalling targets. Our data reveal a molecular machinery that controls the photoperiodic regulation of flowering and growth and offer insight into how plants adapt to seasonal changes.

[1] Ghent University, Department of Plant Biotechnology and Bioinformatics, B-9052 Ghent, Belgium. [2] VIB Center for Plant Systems Biology, B-9052 Gent, Belgium. [3] Department of Molecular Genetics, Ohio State University, Columbus, Ohio 43210, USA. [4] Department of Plant Biology, UC Davis, Davis, California 95616, USA. [5] Department of Medical Protein Research, VIB, B-9000 Ghent, Belgium. [6] Department of Biochemistry, Ghent University, B-9000 Ghent, Belgium. * These authors contributed equally to this work. † Present address: Laboratoire de Biologie Computationnelle et Quantitative, Sorbonne Universités, UPMC, Institut de Biologie Paris-Seine, CNRS, 75005 Paris, France. Correspondence and requests for materials should be addressed to A.G. (email: alain.goossens@ugent.vib.be).

To survive, plants must efficiently 'tell time' to predict variations of an ever-changing environment. In temperate and polar regions, plants established remarkable physiological adaptations to seasons, including the transition to flowering or the modulation of growth. These mechanisms are greatly influenced by the day length (photoperiod), which is sensed in plants through a coordinated system involving external light cues and the internal circadian clock[1,2]. The *Arabidopsis* core oscillator is composed of a set of interlocked feedback loops relaying at different times of the day, with daytime loops repressing evening components and vice versa[3]. In the early evening, the clock components EARLY FLOWERING3 (ELF3), ELF4 and LUX ARRHYTHMO (LUX) are expressed to assemble the Evening Complex (EC). The EC regulates essential rhythmic processes at this period of the day, such as diurnal hypocotyl growth by repressing the transcription factors (TFs) *PHYTOCHROME INTERACTING FACTOR4* (*PIF4*) and *PIF5* (ref. 4). In *Arabidopsis* diurnal hypocotyl elongation is enhanced in short days (SD) compared to long-day (LD) conditions[5]. During SD, PIF4 and PIF5 peak at dawn, directly activating the expression of genes involved in cell division and expansion, and thus triggering growth at the end of the night[4–7]. The stability of PIF proteins is additionally regulated by mechanisms relaying on phytohormonal and light signalling pathways. Red light-activated phytochrome B (phyB) triggers PIF4 and PIF5 proteasomal degradation during daytime, whereas gibberellic acid (GA) promotes their action at the end of the night[8–10]. The reduced activity of phyB in SD promotes PIF protein accumulation, which stimulates hypocotyl growth in long nights[11,12]. Another pathway tightly regulated by the interplay between the evening clock and light signalling is that of photoperiodic flowering. *Arabidopsis* flowers earlier with fewer leaves in LD than in SD conditions, in a process that is mediated by light, clock and hormonal pathways[1,13]. ZTL and FKF1 are blue-light photoreceptor ubiquitin ligases that interact with GIGANTEA (GI), a major mediator between the circadian clock and the photoperiodic flowering pathway[11]. The blue light-stabilized FKF1–GI complex will preferentially assemble in LD to regulate flowering time by controlling protein stability of a family of floral repressors called CYCLIN DOF FACTORs (CDFs)[14]. CDF-mediated repression will promote transcription of *CONSTANS* (*CO*), encoding the TF that subsequently activates expression of *FLOWERING LOCUS T* (*FT*)[14–16]. The accumulation of FT constitutes a florigen signal that triggers the transition from vegetative growth to flowering by activating expression of floral identity genes in the shoot apical meristem (SAM) such as of the TF-encoding *APETALA1* (*AP1*)[1,17]. Although the functioning of the clock and its associated input pathways are being elucidated with an increasing pace, knowledge of the mechanisms that mediate the clock output in a seasonal context is still lacking. Recently, the FAR1 RELATED SEQUENCE (FRS) family of TFs emerged in *Arabidopsis* as key regulators of plant development[18]. FAR-RED ELONGATED HYPOCOTHYL3 (FHY3) and its paralogue FAR-RED IMPAIRED RESPONSE1 (FAR1) are the two best-characterized members of the FRS family, being essential transcriptional activators acting downstream of phyA to regulate photomorphogenic development and coordinating the activation of circadian clock evening components[19,20]. Furthermore, FHY3 was recently shown to play an important role in flower development by directly regulating expression of genes involved in floral meristem determinacy[21].

Here, we demonstrate that two hitherto uncharacterized proteins of the FRS family, FRS7 and its paralogue FRS12, are expressed according to circadian and photoperiodic rhythms, and are involved in the regulation of flowering time and growth. Molecular analysis, including profiling of protein–protein and protein–DNA interactions, protein localization, and trans-activation activity, combined with detailed phenotypic profiling of *FRS7* and *FRS12* gain- and loss-of-function lines indicate that these proteins act as negative regulators of flowering and growth, at least in part by binding to the promoters and repressing the expression of *GI* and *PIF4*.

## Results

**Circadian and photoperiodic regulation of *FRS7* and *FRS12*.** Within the FRS family, many members still have unknown functions. In this study we focused on the hitherto uncharacterized proteins FRS7 (AT3G06250) and its paralogue FRS12 (AT5G18960) (Supplementary Fig. 1) and assessed their possible involvement in the regulation of time- and/or light-dependent developmental processes. We first monitored their circadian gene expression dynamics by transiently expressing the firefly luciferase (fLUC) reporter placed under regulation of the *FRS7* and *FRS12* promoters in *Arabidopsis* Col-0 wt protoplasts entrained under a 12L:12D cycle and then transferred to continuous red light. Bioluminescence levels of both reporter constructs showed robust rhythmic expression patterns declining during subjective day, increasing during subjective night to peak near dawn (Fig. 1a and Supplementary Fig. 2). We next analysed the diurnal expression of *FRS7* and *FRS12* in Col-0 wt *Arabidopsis* seedlings growing under SD and LD photoperiods. Transcripts of both genes were accumulating to higher levels throughout the diurnal cycle in SD as compared to LD (Fig. 1b). However, no robust rhythmic expression of either *FRS7* or *FRS12* transcripts could be observed, in either SD or LD diurnal conditions, contrasting with the observed circadian activities of their corresponding promoters.

To determine if FRS7 and FRS12 protein levels oscillate, we generated plants expressing HA-tagged FRS7 and FRS12 under their respective promoters and followed protein abundances in LD and SD growth. Similar to *FRS7* and *FRS12* transcripts, FRS7-HA and FRS12-HA accumulated highly in SD in comparison to LD at all time points assessed (Fig. 1e), thus indicating a photoperiodic regulation of both proteins. Pronounced diurnal protein oscillations were not observed in any of these conditions (Fig. 1c,d). Taken together, these results suggest that circadian and photoperiodic rhythms influence *FRS7* and *FRS12* transcript abundance whereas the protein levels are predominantly influenced by the photoperiod.

**FRS7 and FRS12 redundantly regulate growth and flowering.** Diurnal hypocotyl growth is a rhythmic process controlled by light and the circadian clock[6,12]. Considering the potential functions of FRS proteins as photomorphogenic regulators and the differential accumulation of FRS7 and FRS12 under different light:dark regimes, we inquired if hypocotyl growth was compromised in *frs7-1*, *frs12-1* and *frs7-1;frs12-1* mutants in parallel to lines ectopically overexpressing *FRS7* or *FRS12* growing under different photoperiods. No significant differences with Col-0 wt seedlings were observed either for *frs7-1*, *frs12-1*, *frs7-1;frs12-1* mutants or overexpression lines under LD growth (Fig. 2a). In contrast, single *frs7-1* and double *frs7-1;frs12-1* mutants presented increased hypocotyl elongation in SD compared to Col-0 wt seedlings (Fig. 2a,b). Conversely, SD-grown lines ectopically overexpressing *FRS7* or *FRS12* displayed significantly reduced hypocotyls. To further support the coordinated function of *FRS7* and *FRS12*, we generated a second *frs7;frs12* double mutant (CRISPR #3–11) using clustered, regularly interspaced short palindromic repeats (CRISPR)–CRISPR-associated 9 (Cas9) genome editing (Supplementary Fig. 3a–f). T3 plants homo-allelic for out-of-frame mutations at both loci were analysed for hypocotyl elongation, which indicated

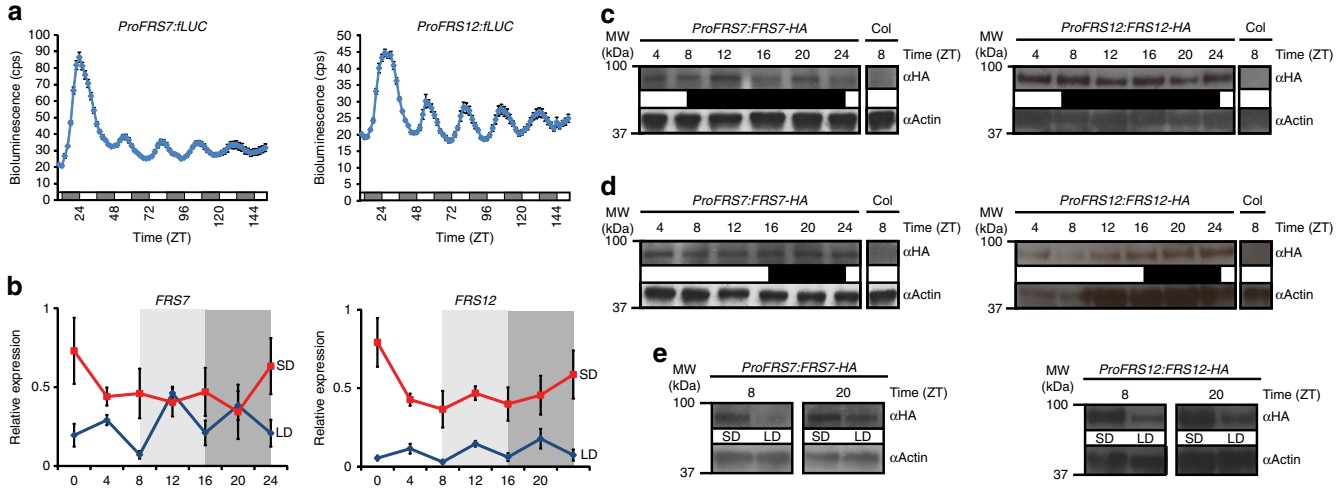

**Figure 1 | FRS7 and FRS12 are circadian and photoperiodic-regulated genes.** (**a**) Circadian bioluminescence expression analysis of *FRS7* and *FRS12*. Four-week-old *Arabidopsis* Col-0 wt protoplast cells from plants growing at 12L:12D cycle were transfected with *ProFRS7:fLUC* or *ProFRS12:fLUC* reporter constructs and transferred to constant red light at ZT9 for image acquisition with 2-h intervals for 1 week. Data represent the mean ± s.e.m. ($n = 6$, corresponding to six wells of protoplasts that were imaged and averaged for each time point of the figure). Two trials were performed, the results of which are shown in this panel and in Supplementary Fig. 2, respectively. White and grey regions indicate subjective light and dark period, respectively. (**b**) Diurnal oscillations of *FRS7* and *FRS12* transcript levels in Col-0 wt seedlings grown in SD or LD light conditions. '1' represents the highest level of expression for a particular gene. Light and dark grey rectangles represent the dark period in SD and LD, respectively. Values represent the average expression of three biological replicates ± s.e.m. (**c,d**) Immunoblot analysis showing the diurnal oscillation patterns of HA-tagged FRS7 and FRS12 proteins expressed through their respective native promoters in SD (**c**) and LD (**d**) growth conditions. White and black regions indicate light and dark period, respectively. (**e**) Comparison between SD and LD accumulation levels of HA-tagged FRS7 and FRS12 proteins expressed under control of their respective native promoters at ZT8 and ZT20. Uncropped versions of all immunoblot images presented in **c**–**e** are shown in Supplementary Fig. 15.

that, as the *frs7-1;frs12-1* mutant, this line presented increased hypocotyl elongation specifically under SD growth (Supplementary Fig. 3g). Taken together, these results account for the coordinated functions of FRS7 and FRS12 to repress hypocotyl growth in a photoperiodic-dependent manner.

Next, we examined spatial expression of *FRS7* and *FRS12* in transgenic plants expressing the β-glucuronidase (GUS) reporter driven by their respective promoters. Both genes co-express in leaf vasculature and SAM, though *FRS7* also expresses in leaf mesophyll cells (Fig. 3a and Supplementary Fig. 4a).

Leaf growth is a rhythmic process that peaks soon after dawn[22]. Because of the leaf-localized expression of *FRS7* and *FRS12*, and the observed phenotypes in hypocotyl growth, we tested the influence of FRS7 and FRS12 on the dynamics of leaf elongation. To this end, we employed a recently developed time-resolved *in vitro* growth imaging system (IGIS)[23], accompanied with detailed leaf area measurements to study rosette growth for 21 days in lines in which the *FRS7* and *FRS12* functions had been altered. Rosette growth of the double *frs7-1;frs12-1* mutant was significantly increased by 23.3% compared to Col-0 wt plants. In contrast, *Pro35S:FRS7-HA-1* and *Pro35S:FRS12-HA-1* lines were severely reduced in growth, by 27.5% and 64.2%, respectively, compared to Col-0 wt plants (Supplementary Fig. 5). To test if FRS7 and FRS12 affect leaf growth according to the photoperiod, we then measured total leaf areas during early development of plants with altered expression of *FRS7* and/or *FRS12* grown under SD or LD. Similar results were obtained under both photoperiods. Increased variability was observed in SD, which may be due to the fact that plants grow slower in SD[24] (Fig. 3b,c and Supplementary Table 1) and differences in rosette areas become more important only at later developmental stages (after 30 days) (Fig. 3d). Because at later stages other developmental factors, such as flowering and leaf senescence, make accurate comparisons of leaf areas technically difficult, this

was not further assessed. Nonetheless, in our setup, the double *frs7-1;frs12-1* mutant showed enhanced leaf areas compared to Col-0 wt plants in both photoperiodic conditions. In contrast, both *Pro35S:FRS7-HA-1* and *Pro35S:FRS12-HA-1* lines showed considerable reductions in leaf areas compared to Col-0 wt. Taken together, these results suggest that FRS7 and FRS12 are involved in the modulation of rosette leaf growth.

Seasonal flowering is regulated by the photoperiod-sensing pathway that expresses distinctively at the leaf vasculature[25–27]. Therefore, we also examined if *FRS7*- and *FRS12*-altered lines presented impaired flowering times in LD or SD. Plants of the single *frs7-1* mutant flowered 2 days earlier than Col-0 wt plants in LD with ∼1 leaf less and 8 days earlier but with a similar number of leaves compared to Col-0 wt plants in SD (Fig. 4a–d and Supplementary Fig. 4b). No significant differences in flowering time were observed in plants of the single *frs12-1* mutant compared to Col-0 wt in both light:dark regimes. Plants of the double *frs7-1;frs12-1* mutant flowered 2 days earlier than Col-0 wt in LD with ∼2 leaves less and flowered dramatically premature under SD, that is, with ∼24 leaves less and ∼19 days earlier compared to Col-0 wt (Fig. 4a–e). In contrast, lines ectopically expressing *FRS7* and *FRS12* flowered significantly later than Col-0 wt in LD, that is, with 7 leaves more and 10 days later compared to Col-0 wt (Supplementary Fig. 4c). Likewise, under SD conditions, these transgenic lines flowered significantly later than Col-0 wt (Supplementary Fig. 4d). Altogether, these findings support cooperative functions of FRS7 and FRS12 in the regulation of flowering time.

**FRS7 and FRS12 integrate a repressor complex in the nucleus.** To explore the functioning of the two FRS proteins at the protein–protein interaction level, we used an advanced tandem affinity purification–mass spectrometry (TAP–MS) method with

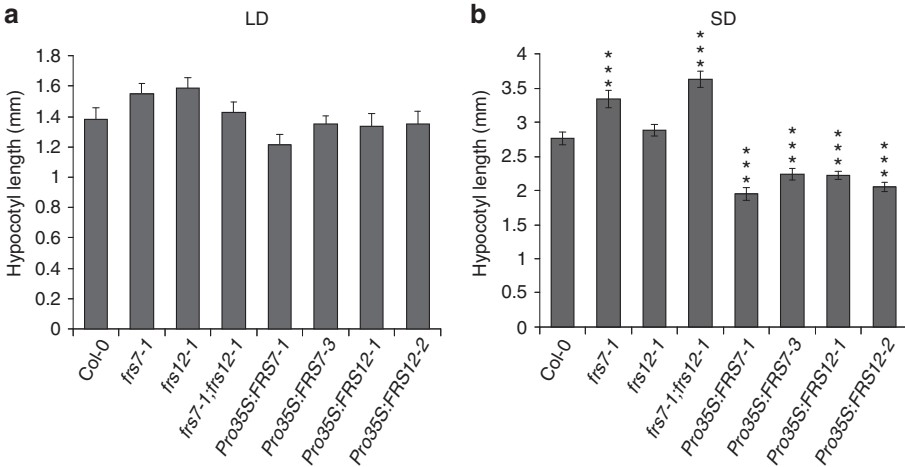

**Figure 2 | FRS7 and FRS12 repress hypocotyl growth in a photoperiodic-dependent manner.** Hypocotyl length measurements of *Arabidopsis* Col-0 wt seedlings compared to gain- and loss-of-function lines of *FRS7* and *FRS12* grown for 10 days in LD (**a**) and SD (**b**). Values represent the average of at least 20 biological replicates ± s.e.m. in SD and 16 in LD conditions (***$P < 0.001$, *t* test).

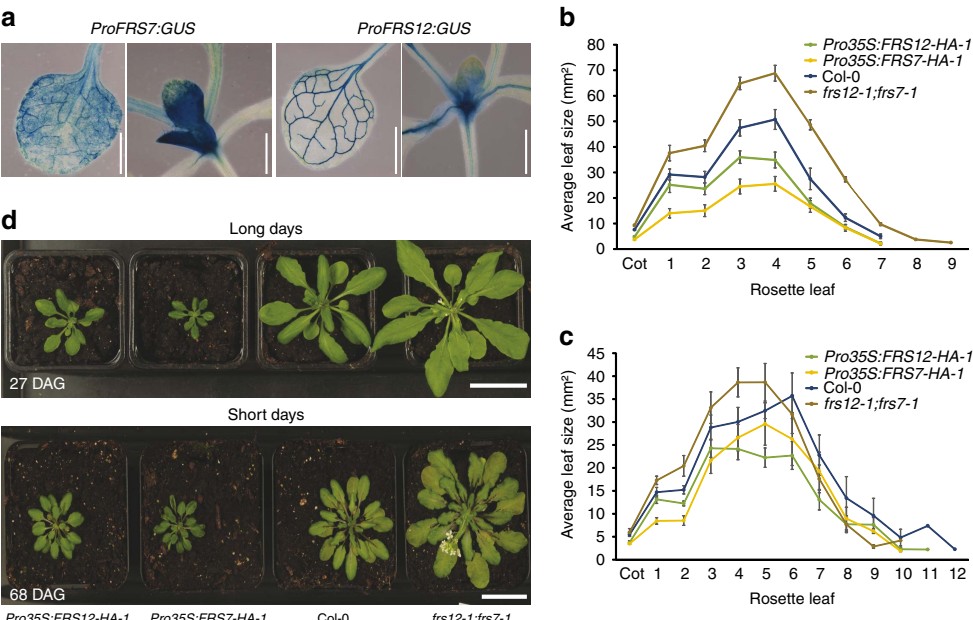

**Figure 3 | FRS7 and FRS12 modulate leaf-rosette growth.** (**a**) GUS histochemical analysis of the spatial expression of *FRS7* and *FRS12* in 14-day-old seedlings. Scale bars, 1 mm. (**b**,**c**) Leaf series of Col-0 wt, *frs7-1;frs12-1*, *Pro35S:FRS7-HA-1* and *Pro35S:FRS12-HA-1* lines grown under LD (**b**) or SD (**c**) conditions for 18 and 30 days, respectively. Values represent the average of eight biological replicates ± s.e.m. (**d**) Representative photographs comparing Col-0 wt to *frs7-1;frs12-1* mutant plants and *FRS7* and *FRS12* overexpressing lines under LD or SD conditions for 27 and 68 days, respectively (DAG: days after germination). Scale bars, 3 cm.

improved sensitivity[28]. First, C-terminally TAP-tagged FRS12 was expressed constitutively and purified first from *Arabidopsis* cell cultures growing in absence of circadian or photoperiodic cues, that is, under continuous darkness (cD). This method identified FRS7, the uncharacterized linker histone-like protein HON4, the AT-HOOK MOTIF NUCLEAR LOCALIZED PROTEIN9 (AHL9) and AHL14 as *bona fide* interactors of FRS12 (Fig. 5a, Supplementary Table 2 and Supplementary Data 1). Next, we examined if the FRS7–FRS12 complex could also assemble in LD, despite the low expression of these proteins under this photoperiod (Fig. 1e). To this end, TAP-tagged FRS12 was purified from cell cultures entrained under LD and collected at day (ZT4) and night (ZT20) time periods. FRS7 was identified in

all tested conditions, supporting the assumption that FRS7 and FRS12 assemble and that the corresponding complex is also active in LD photoperiods (Fig. 5a and Supplementary Data 1). Furthermore, HON4 was also recruited by FRS12 in LD but only at night-time (ZT20), whereas AHL14 co-purified only at daytime (ZT4). These results suggest that the FRS7–FRS12 complex assembles at day- and night-times, and may recruit additional or distinct partners over the course of the diurnal cycle.

Confocal microscope imaging of transgenic *Arabidopsis* seedlings ectopically expressing FRS7-GFP and FRS12-GFP showed an exclusive nuclear localization of both proteins (Supplementary Fig. 6a). Accordingly, bimolecular fluorescence complementation (BiFC) assays in *Nicotiana benthamiana* leaves

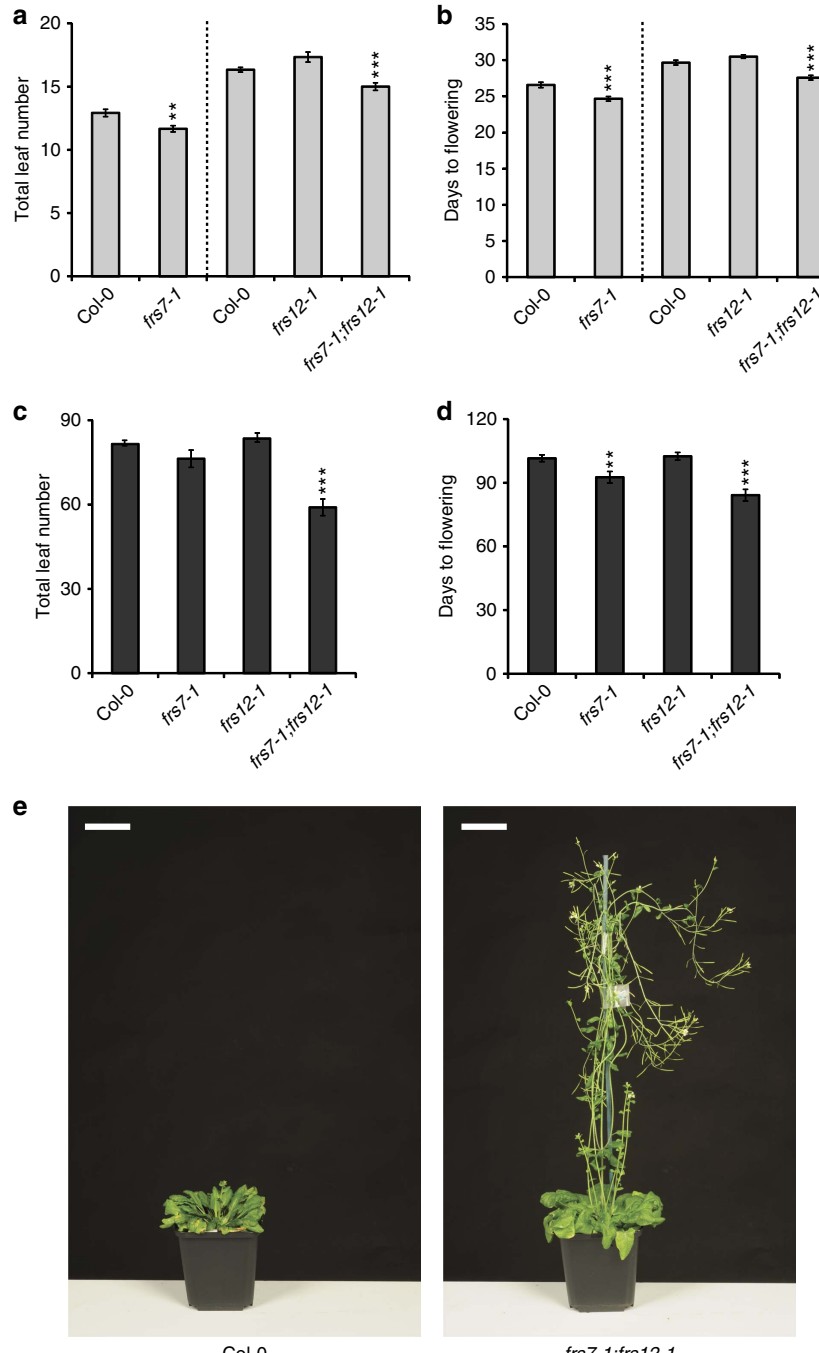

**Figure 4 | FRS7 and FRS12 modulate flowering time.** (**a**,**b**) Flowering time measurements as total leaf number (**a**) and days to flowering (**b**) of LD-grown Col-0 wt *Arabidopsis* plants compared to loss-of-function lines of *FRS7* or *FRS12*. (**c**,**d**) Flowering time measurements as total leaf number (**c**) and days to flowering (**d**) of SD-grown Col-0 wt *Arabidopsis* plants compared to loss-of-function lines of *FRS7* or *FRS12*. Values represent the average of 12 biological replicates ± s.e.m.; **$P < 0.01$, ***$P < 0.001$, $t$ test. (**e**) Representative photographs comparing a *frs7-1;frs12-1* double mutant to a Col-0 wt plant flowering under SD growth. Scale bars, 5 cm.

showed a nuclear-localized GFP signal when co-expressing all combinations of nGFP-tagged FRS7 or FRS12 with cGFP-tagged FRS7 or FRS12, corroborating *in vivo* nuclear interactions (Fig. 5b and Supplementary Fig. 6c). A nuclear GFP signal was also observed when co-expressing cGFP-tagged FRS7 or FRS12 with nGFP-tagged HON4 or AHL14, confirming the nuclear-localized interaction of these proteins (Supplementary Fig. 6b,c). Transient trans-activation assays in tobacco protoplasts determined that both FRS7 and FRS12 could repress *fLUC*

reporter gene expression, defining these proteins as transcriptional repressors (Fig. 5c). Taken together, these results designate FRS7 and FRS12 as part of a transcriptional repressor complex.

To further investigate the molecular functions of the FRS7–FRS12 complex, we generated transgenic inducible *Arabidopsis* lines producing FRS12 fused to the hormone binding domain of the rat glucocorticoid receptor (GR), which translocates to the nucleus in the presence of dexamethasone (DEX). LD-grown seedlings of *Pro35S:FRS12-GR-1* and *Pro35S:GFP-GR*

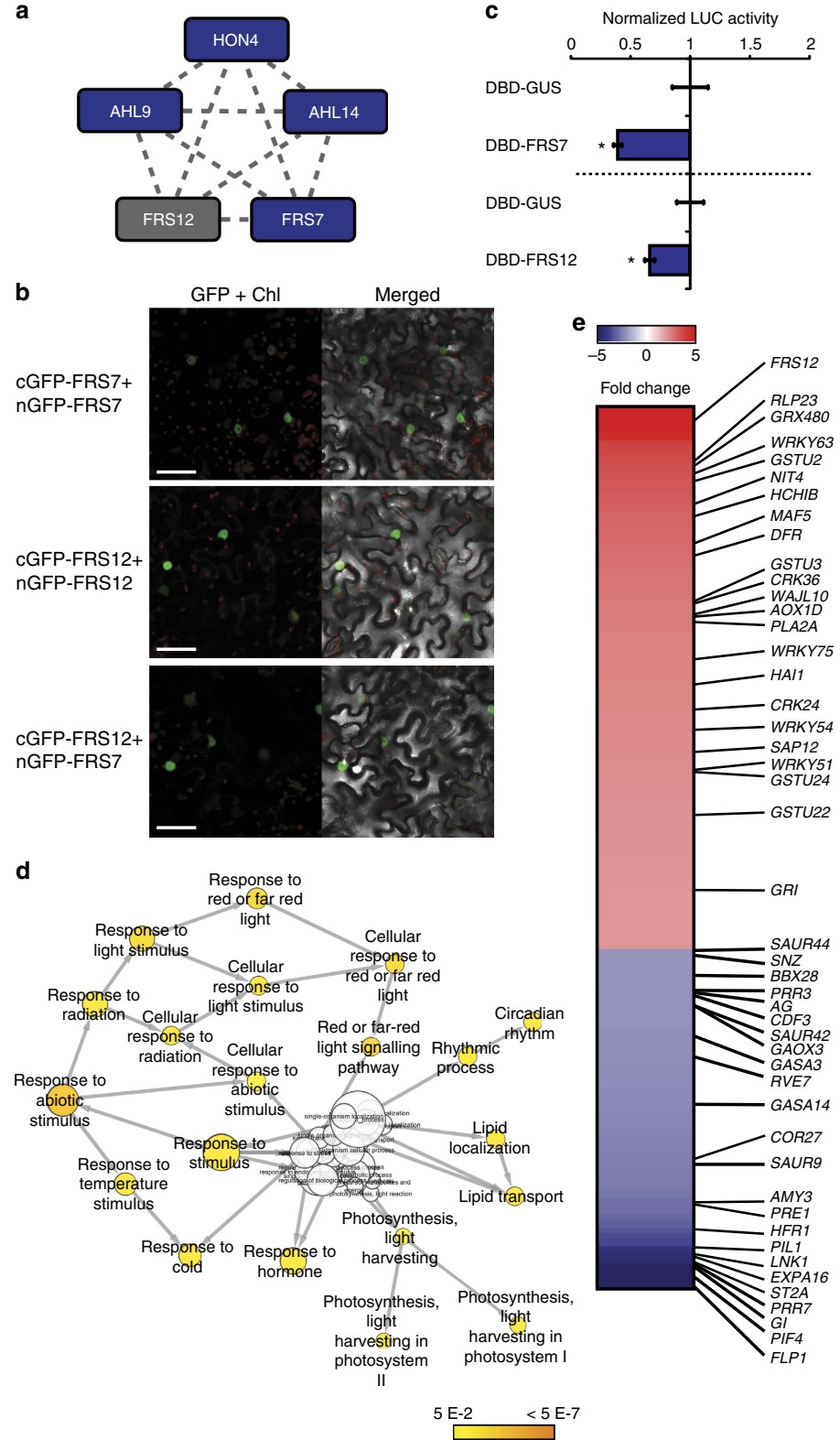

**Figure 5 | FRS7 and FRS12 integrate a transcriptional repressor complex. (a)** Spoke-model of the protein interaction network identified for FRS12 by TAP–MS. The dashed edges represent previously unknown interactions. Blue nodes represent preys and the grey node represents the bait protein. **(b)** BiFC analysis in *N. benthamiana* leaves of FRS7 and FRS12 nuclear interactions. Scale bars, 50 μm. **(c)** Transient trans-activation assay in tobacco protoplasts co-transfected with a *ProUAS:fLUC* reporter construct, effector constructs (*Pro35S:DBD-FRS12* and *Pro35S:DBD-FRS7*) and a *Pro35S:rLUC* construct to normalize expression. Mean expression values ± s.e.m. are depicted relative to the respective *Pro35S:GUS-DBD* control effector construct; *n* = 8, \**P* < 0.05, *t* test. **(d)** GO-enrichment network of the genes identified by RNA-Seq as repressed (fold change ≥ 2) in the *Pro35S:FRS12-GR-1* line. **(e)** Heat map representing the genes identified by RNA-Seq as differentially regulated (fold change ≥ 2) in the *Pro35S:FRS12-GR-1* line. Indicated genes are associated to stress responses, flowering time, light signalling, growth or circadian rhythms.

(control) lines were treated with DEX and harvested 4 h after treatment. The RNA-Seq transcriptome comparison between these lines revealed 217 upregulated and 136 downregulated genes with a fold change >2 (Supplementary Data 2). GO-enrichment analysis indicated that *FRS12* overexpression resulted in the upregulation of a number of stress-related genes (Supplementary Data 3). These genes also grouped under phytohormone-responsive terms related to stress responses such as jasmonate, ethylene or salicylic acid. A set of specialized (also called secondary) metabolism genes such as those from the camalexin biosynthesis pathway were also enriched in the *Pro35S:FRS12-GR1* line. However, considering the repressor activity of the FRS7–FRS12 complex, we focused in this study on the downregulated genes, for which GO enrichment highlighted circadian clock, photoperiodic processes and red-light signalling amongst the most significantly enriched terms (Fig. 5d and Supplementary Data 3). In relation to diurnal growth, *PIF4* displayed a more than 10-fold downregulation and represented the fifth most highly repressed gene of this analysis (Fig. 5e and Supplementary Data 3). Furthermore, induced ectopic *FRS12* overexpression also resulted in repression of several known PIF4-upregulated target genes[29], including the leaf expansion regulator *GIBBERELLIN-STIMULATED ARABIDOPSIS14* (*GASA14*)[30] and the growth-related TFs, *PHYTOCHROME INTERACTING FACTOR3-LIKE 1* (*PIL1*), *LONG HYPOCOTYL IN FAR-RED1* (*HFR1*)[31] and *PACLOBUTRAZOL RESISTANCE1* (*PRE1*)[32]. In relation to photoperiodic flowering time, *GI* also featured amongst the most downregulated genes in the induced *Pro35S:FRS12-GR-1* line (Fig. 5e). Furthermore, our transcriptome analysis identified other repressed genes with known functions in flowering, such as the photoperiod-sensing pathway constituent *CYCLIN DOF FACTOR3* (*CDF3*), *SCHNARCHZAPFEN* (*SNZ*)[33], the *FPF1-LIKE PROTEIN1* (*FLP1*) or the floral meristem determinacy modulator *AGAMOUS* (*AG*)[34]. Finally, also genes encoding circadian clock components were found to be repressed in this line. In addition to *GI*, these included the *PSEUDO RESPONSE REGULATOR7* (*PRR7*), *PRR3* and the evening time co-activator *NIGHT LIGHT-INDUCIBLE AND CLOCK-REGULATED1* (*LNK1*)[3,35]. The expression patterns observed in the RNA-Seq analysis were corroborated for *PIF4*, *PIL1*, *PRR7*, *GI* and *FLP1* by qPCR analysis in two independent *Pro35S:FRS12-GR* lines. This analysis confirmed the accuracy of the RNA-Seq analysis and demonstrated the repressing effect of ectopic *FRS12* expression (Supplementary Fig. 7).

**FRS7–FRS12 repress photoperiodic flowering and diurnal growth.** To define genomic targets of the FRS7–FRS12 complex, we performed tandem chromatin affinity purification in *Arabidopsis* cells ectopically expressing tagged *FRS12*, followed by next-generation sequencing analysis (TChAP-Seq). The overlap between the two replicates consisted of 2,743 shared bound loci (Supplementary Data 4 and 5). GOslim analysis of the bound loci highlighted a significant enrichment in several biological processes including signal transduction, cell differentiation and flower development (Supplementary Fig. 8a). Next, we compared the FRS12-bound genes to publicly available genome-wide chromatin immunoprecipitation (ChIP) experiments for 27 TFs and constructed a comparative TF co-binding matrix[36]. Remarkably, the floral meristem determinacy transcription factors AP1, AP2, SEPALLATA3 (SEP3), AGAMOUS-LIKE15 (AGL-15) and the circadian growth regulator PIF4 were included amongst the five most significant TFs sharing significant co-bound genes with FRS12 (Supplementary Fig. 8b,c). This overlap consisted for PIF4 of 519 genes ($\sim\frac{1}{4}$ of PIF4-bound

genes) and for AP1 of 991 genes ($\sim\frac{1}{4}$ of AP1-bound genes), suggesting that FRS12 potentially represses large portions of PIF4 and AP1 (which is downstream of GI) gene targets.

The comparison between the TChAP-Seq and the RNA-Seq datasets allowed pinpointing those genes that were both bound and regulated by FRS12. A total of 42 genes obeyed to these two criteria, corresponding to 12% of the genes detected as significantly altered in the RNA-Seq analysis (Fig. 6a and Supplementary Table 3). Consistent with our co-binding comparisons, FRS12 bound to and repressed many target promoters of PIF4 and circadian clock TFs, designating FRS12 as a direct modulator of diurnal growth (Supplementary Fig. 9). The promoters of *PIF4* and *PIL1* illustrate diurnal growth targets of FRS12 (Fig. 6b). The *GI* promoter was also bound and repressed by FRS12, supporting its function as direct repressor of the photoperiod-sensing pathway (Fig. 6b). Subsequently, we inspected enriched bound sites of FRS12 through *de novo* motif analysis using the peak-motifs tool[37]. This analysis identified three motifs, that is, TGTGTG, TATATATATATATATAT and TATACATA hereafter called FRS12-Box1 (FRB1), FRB2 and FRB3, respectively (Fig. 6c). These motifs are centred on the summits of the corresponding binding peaks, therefore, establishing a highly significant overlap between FRS12 binding and these motifs (Supplementary Fig. 8d). The FRB1 was found in peak summits of the *PIF4* and *GI* promoters, whereas FRB2 and FRB3 were found in peaks of the *PIL1* promoter (Fig. 6b). Recently, three evolutionary conserved regions of the *GI* promoter, named CRM1, 2 and 3, were found important for the circadian and light regulation of *GI*[38]. Furthermore, three evening elements (EEs) present in the CRM2 region were defined as essential for the evening-expression pattern of this gene[38]. Notably, two of the FRS12-binding peaks in the *GI* promoter corresponded exactly to the CRM1 and CRM2 regions. Sequence analysis in CRM2 showed that the FRB1 motif partially overlaps with one EE and is flanked by the two other EEs. This result suggests that FRS7–FRS12 could repress *GI* expression by physical occupation of evening clock activators' sites in the *GI* promoter (Fig. 6b). To confirm the binding of FRS12 to the promoters of *PIF4*, *PIL1* and *GI*, ChIP-qPCR analysis was carried out using the *ProFRS12:FRS12-HA* line grown in SD and harvested at ZT8. The analysed regions consisted in the FRB1 motif present in the CRM2 region for *ProGI*, the peak summit of the most highly bound region for *ProPIF4*, and the peak summit containing the FRB2 and FRB3 motifs for *ProPIL1*, respectively (Fig. 6b). The *ProFRS12:FRS12-HA* line showed enriched binding compared to Col-0 wt seedlings in the analysed regions of all three promoters (Fig. 6d). On the contrary, no enrichment differences were observed when amplifying an *ACTIN2* region as control for unbound DNA (Fig. 6d).

Because the FRS7–FRS12 complex accumulates preferentially in SD, we questioned if FRS12 could show altered binding to its targets in a circadian and/or photoperiodic-dependent manner. Therefore, we grew the *ProFRS12:FRS12-HA* line in SD and LD conditions, harvested samples at ZT4 or ZT20 time points and performed ChIP-qPCR. Enriched binding of FRS12 to *GI* and *PIF4* promoters was observed only under SD conditions; in the case of *ProGI* only during daytime (ZT4) and of *ProPIF4* both during day- (ZT4) and night-times (ZT20) (Supplementary Fig. 10). These results support the idea that the FRS7–FRS12 complex is more active under SD conditions and regulates its target promoters in a photoperiodic-dependent manner, which is in agreement with for instance the SD-specific hypocotyl elongation phenotypes.

To further examine if FRS7 and FRS12 could modulate the activation of the *PIF4*, *PIL1* and *GI* promoters, we performed transient expression assays in leaves of SD-entrained *N. benthamiana* plants. Significant reduction in the *ProPIF4-min35S:fLUC* reporter

activity was observed when co-expressing *FRS7*, *FRS12* or *FRS7* together with *FRS12*, indicating repressor activity for these proteins on this promoter (Fig. 6e). Because the FRS7–FRS12 complex co-

binds and represses targets of PIF4, we examined if this complex could compete with PIF4 to modulate the activation of their targets. We tested this hypothesis in a transient expression assay using the

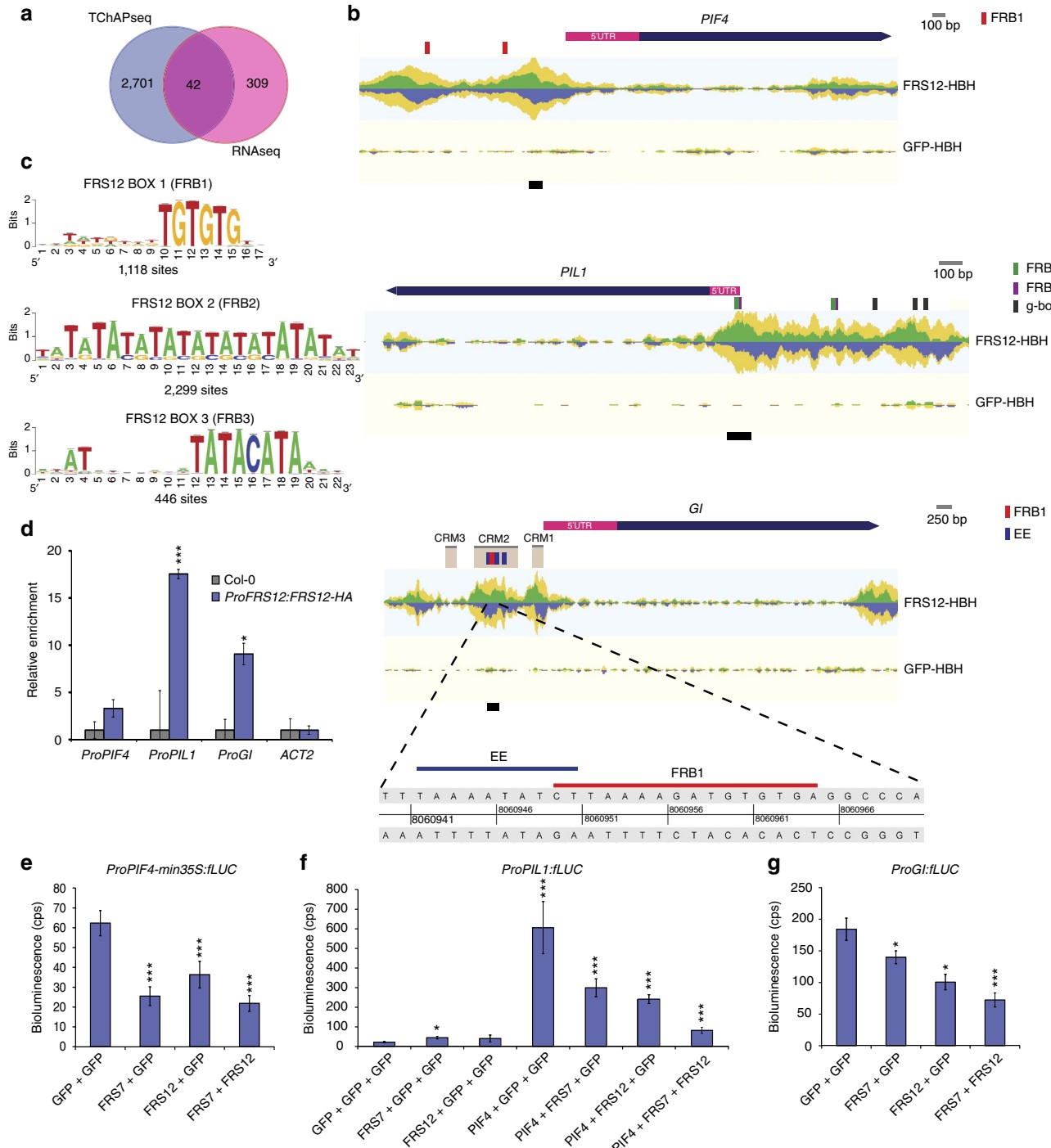

**Figure 6 | The FRS7–FRS12 complex binds and represses genes responsible for diurnal growth and flowering.** (**a**) Venn diagram comparing the FRS12 physically bound genes to the transcriptionally regulated genes in the *Pro35S:FRS12-GR-1* line. (**b**) Representations of the TChAP-Seq FRS12 binding peaks located at the *PIF4*, *PIL1* and *GI* promoters. The reads are piled up in forward reads above the axis in green and reverse reads below the axis in blue. Total coverage is indicated in yellow. Coloured bars on top of promoters designate protein–DNA-binding motifs. Black lines below promoters highlight selected regions for ChIP-qPCR analysis. (**c**) Logo representation of the FRS12 DNA-binding elements FRB1, FRB2 and FRB3, identified by *de novo* motif enrichment analysis. (**d**) ChIP-qPCR assay of selected fragments in the *PIF4*, *PIL1* and *GI* promoters. A transgenic *Arabidopsis* line expressing *ProFRS12:FRS12-HA* was grown for 10 days in SD conditions and harvested at ZT8 for analysis. Enrichment values were normalized to respective inputs and represented relative to Col-0 wt plants (background control). Values represent the mean of three biological replicates ± s.e.m.; *$P < 0.05$, ***$P < 0.001$, *t* test. (**e–g**) Transient expression assays in *N. benthamiana* showing the trans-repression of the *PIF4*, *PIL1* and *GI* promoters by FRS7 and FRS12. Values represent the mean of eight technical replicates ± s.e.m.; *$P < 0.05$, ***$P < 0.001$, *t* test.

                                    

*ProPIL1:fLUC* construct as reporter of PIF4 activity[39]. As expected, PIF4 trans-activation resulted in a 30-fold activation of the reporter compared to the GFP control (Fig. 6f). Co-expression of *PIF4* with *FRS7* or *FRS12* led to a 50% decrease in the fLUC reporter activity and the combination of *FRS7*, *FRS12* and *PIF4* led to a reduction of 86% of the reporter's activation. This result demonstrates that FRS7 and FRS12 function additively to weaken PIF4 activity when these proteins co-bind to promoters of their target genes. Finally, both FRS7 and FRS12 significantly reduced the *ProGI:fLUC* reporter activity, by 25% and 50%, respectively, and the combined expression of both FRS proteins led to a reduction of 60% compared to the GFP control, further supporting their additive repressive functions (Fig. 6g). Taken together, these results define the FRS7–FRS12 complex as a molecular machinery that represses diurnal growth and photoperiodic flowering through direct repression of the *PIF4* and *GI* gene networks.

Given that *PIF4*, *GI* and their respective gene networks rhythmically oscillate, we assessed whether the *frs7-1;frs12-1* mutant and ectopically *FRS7-* and *FRS12* overexpressing lines would present altered expressions of these genes during the diurnal cycle in LD and/or SD photoperiods. Time-course studies showed rhythmic expression peaks with increased amplitudes at dusk (ZT8) for *PIF4* in the *frs7-1;frs12-1* line compared to Col-0 wt plants in SD growth (Supplementary Figs 11 and 12). Conversely, repressed *PIF4*-expression amplitudes were observed in the *FRS7* and *FRS12* overexpressing lines under SD growth. Furthermore, ectopic overexpression of *FRS7* or *FRS12* induced marked reductions in the expression amplitudes of the PIF4 targets *PIL1* or *HFR1* compared to Col-0 wt plants, mainly under SD growth (Supplementary Fig. 11). Opposed to this observation, the *frs7-1;frs12-1* line did not show marked changes in expression of the latter two genes, which may be the result of the genetic redundancies between PIF4 and its homologues[40].

No significant changes in the expression levels of *GI* and the photoperiodic flowering pathway components *FKF1* and *CO* were observed in the *frs7-1;frs12-1* mutant compared to Col-0 wt under both SD and LD photoperiods (Supplementary Fig. 11). However, ectopic overexpression of *FRS7* or *FRS12* repressed the expression amplitudes of *GI* under both SD and LD growth. Taken together, our results indicate that the FRS7–FRS12 complex represses *GI*, *PIF4* and PIF4-downstream target genes.

Enhanced expression of *PIF4* was only observed in the *frs7-1;frs12-1* double mutant but not in the single *frs7-1* or *frs12-1* mutants, suggesting the redundant functions of these proteins (Supplementary Fig. 13). Given the weak and variable effect on *PIF4* and *GI* expression but the clear effect on flowering caused by loss of *FRS7* and *FRS12* function, FRS7 and FRS12 may modulate flowering time through other, yet unknown, targets than PIF4 and GI (Supplementary Figs 11 and 12). Furthermore, the *frs7-1;frs12-1* line showed no differences in the expression of circadian clock genes, such as *LATE ELONGATED HYPOCOTYL* (*LHY*), *PRR7* and *TIMING OF CAB EXPRESSION 1* (*TOC1*), compared to Col-0 wt plants, which places the functions of the FRS7–FRS12 complex on the output side of the circadian clock (Supplementary Fig. 14). Taken together, our results define the FRS7–FRS12 complex as a photoperiodic- and circadian regulated machinery that modulates essential components of flowering and growth pathways in *Arabidopsis* (Fig. 7).

## Discussion

Our results characterize the FRS7–FRS12 complex as a machinery being regulated according to circadian and photoperiodic rhythms. Components of this complex particularly accumulate under SD and repress circadian clock output targets. Correct timing of flowering

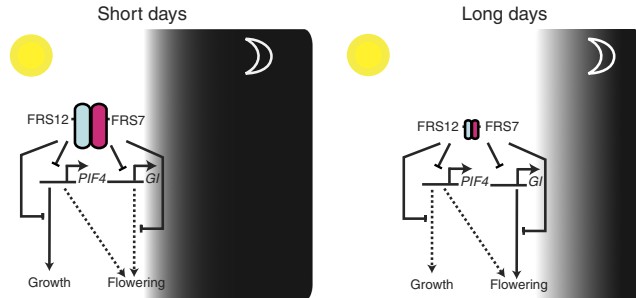

**Figure 7 | The FRS7–FRS12 complex regulates diurnal growth and flowering pathways by repressing expression of *PIF4* and *GI*.** Proposed model of the function of the FRS7–FRS12 complex. FRS7 and FRS12 accumulate more abundantly in SD, bind promoter regions and repress the expression of the *PIF4* and *GI* gene networks to modulate diurnal growth and flowering.

and growth are central events in the life cycle of plants. These processes are coordinated with seasonal changes to ensure an adapted response to the environment and thereby safeguarding fitness. Flowering time is a process tightly dependent on the photoperiod in *A. thaliana*. LD conditions trigger rapid transition from the vegetative to the reproductive stage in this plant[13]. In contrast, this process is considerably delayed in SD. Our data reveal that loss-of-function of the FRS7–FRS12 complex accelerates flowering time independently of the photoperiod, though more strongly in SD.

Similarly to photoperiod-sensing pathway components[25,26], *FRS7* and *FRS12* expression localize at the leaf vasculature and the SAM. These expression patterns together with their increased protein accumulation and binding activity in SD support the functions of the FRS7–FRS12 complex as a regulator of the photoperiod-sensing pathway. Furthermore, the moment to flower is controlled in *Arabidopsis* by the coincidence of an internal rhythm established by the circadian clock together with external day-length information[1]. The core regulation of the photoperiod-sensing pathway is represented by the GI–FKF1 module[1,13]. In LD, accumulation of these proteins overlaps in the late afternoon, allowing the assembly of the GI–FKF1 complex to trigger flowering[2,14,15]. However in SD, the GI protein peaks at dusk whereas FKF1 peaks also once more after dark, which prevents the assembly of the complex and thereby halts flowering. The asynchrony and amplitude of these proteins' expression play a crucial role in the repression of flowering in SD. Accordingly, we found that the FRS7–FRS12 complex could partially regulate this circuit by binding to the promoter of *GI* and repressing its expression under SD photoperiods.

Another mechanism of flowering time regulation is specified by the thermosensory pathway, in which PIF4 accelerates transition to the reproductive stage at high temperatures[41,42]. Considering these functions, the misregulation of PIF4 may also partially contribute to the observed flowering time phenotypes in *FRS7*- and *FRS12*-altered lines. In this context, the repression of *PIF4* by the FRS7–FRS12 complex could constitute an additional layer of regulation to correctly time the transition from the vegetative to the reproductive stage in *Arabidopsis*.

Hypocotyl growth is another key process regulated in a circadian and photoperiodic-dependent manner[4,5,43]. *Arabidopsis* seedlings promote hypocotyl elongation predominantly at the end of the night, around dawn, in SD[5,6]. Similar to photoperiodic flowering, this process is regulated by the coincidence of an internal rhythm given by the circadian clock together with external light cues coming from the environment[2]. In SD, the EC

assembles in the early evening to repress expression of targets such as *PIF4* and *PIF5*, halting growth at this time of the day[2]. However, *PIF4* and *PIF5* are transcribed at the end of the night in SD, creating conditions under which hypocotyl growth is activated[5,11]. Our data reveal functions of FRS7–FRS12 in the regulation of this process as well. Indeed, the FRS7–FRS12 complex acts by binding to the *PIF4* promoter to repress its transcription, and, eventually, as well that of several of its downstream targets, therefore, modulating growth at this time of the day, particularly in SD. Although the EC and FRS7–FRS12 complex share similar functions, our results suggest that these two mechanisms play complementary rather than redundant roles. Mutations affecting components of the EC impair clock function, as observed at the molecular level, that is, by arrhythmic expression of clock components[44]. In contrast, mutations affecting FRS7–FRS12 complex components do not affect the rhythmicity of the clock, placing this complex at the output side of the clock. Finally, the fact that the FRS7–FRS12 complex modulates diurnal growth and photoperiodic flowering without affecting the clock functions might provide useful insights into how to design plants with increased vegetative growth and accelerated flowering time.

## Methods

**Cloning and transformation of plants.** All *A. thaliana* plants used in this study were in the Col-0 ecotype background and all stable plant transformations were carried out by floral dip using *Agrobacterium tumefaciens* strain C58C1. The *frs12-1* mutant in the Col-0 background ecotype was obtained from the SALK T-DNA collection (http://signal.salk.edu), corresponding to the SALK_030182 accession. The initial *frs7-1* mutant was in the Ws-2 background ecotype and was obtained from the INRA T-DNA insertion collection (http://www-ijpb.versaille-s.inra.fr/en/plateformes/cra/index.html), corresponding to the FLAG_196C09 accession. The single *frs7-1* and double *frs7-1;frs12-1* mutant in the Col-0 ecotype were obtained by crossing the parental lines, followed by backcrossing the progeny with Col-0 background for four generations.

To generate the CRISPR line, specific guide sequences targeting the 5′ of the *FRS* genes (GN19-NGG type) were selected using CRISPR-P (http://cbi.hzau.edu.cn/cgi-bin/CRISPR)[45] (Supplementary Fig. 3a) taking into account predicted-single guide RNA (sgRNA) efficiencies using sgRNAscorer (https://crispr.med.harvard.edu/sgRNAScorer/)[46].

An updated overview of estimated sgRNA parameters by CRISP-OR (http://crispor.tefor.net/)[47] can be found in Supplementary Table 7.

The CRISPR vector pDE-Cas9 (Km) was derived from pDE-Cas9 (ref. 48) by replacing the basta resistance cassette with an *nptII* module. The *NOSp-NptII-NOSt* cassette (NptII cassette) was amplified using primers NptII.For and NptII.Rev (Supplementary Table 4). The pDe-Cas9 vector was digested with AatII and PmeI to remove the PPT cassette and the NptII cassette was introduced by in-fusion reaction (Takara Bio USA). The vector contains *A. thaliana* codon-optimized Cas9 under control of the *Petroselinum crispum* ubiquitin promoter (PcUbi4-2) and a gateway recombination cassette to clone sgRNA modules[48]. To allow combining two sgRNA modules, primers (Supplementary Table 4) were designed to amplify the sgRNA cassette from pEn-C1.1 (L1-L2 (ref. 48)) adding appropriate *attB/attBr* flanking sites (B1-B5r and B5-B2) to each fragment. The amplified fragment was then cloned into the corresponding pDONR221 vector (pDONR221 P1-P5r and P5-P2) by Gateway BP reaction to generate entry clones suitable for MultiSite Gateway LR cloning. An additional BbsI site in the pDONR backbone was eliminated by site directed mutagenesis using an In-Fusion reaction using primers noBbsI_F and noBbsI_R (Supplementary Table 4).

For each guide sequence, two complementary 23-bp oligos with 4 bp overhangs (Supplementary Table 4) were annealed and inserted via a cut-ligation reaction with BbsI (Thermo) and T4 DNA ligase (Thermo) in either pMR217 (L1-R5) for *FRS12* or pMR218 (L5-L2) for *FRS7*. The 5′ overhang contains the G initiation nucleotide for the AtU6-26 polIII promoter. Using a MultiSite Gateway LR reaction, both sgRNA modules were then combined with pDE-Cas9 (Km) to yield the final expression clone.

The expression clone was transformed in *A. tumefaciens* C58C1 (pMP90) and used for *Arabidopsis* Col-0 floral dip transformation. Ten primary transformants (T1) were selected on kanamycin and genomic DNA extracted from a seedling leaf. For each gene, a 600–700 bp genomic region of spanning the predicted Cas9 cut site was amplified (Supplementary Table 4) and the amplicon sequenced by standard capillary sequencing. Resulting quantitative sequence trace data was decomposed using TIDE (https://tide.nki.nl/)[49] resulting in an estimation of editing efficiency and identify the dominant indel types. T1 plants were typically chimeric, but with a high editing efficiency (Supplementary Fig. 3b). After selfing, T2 plants were checked for segregation of the T-DNA locus using kanamycin resistance.

Twelve T2 plants from three lines with one T-DNA locus were genotyped using Cas9 specific primers to identify null segregants (Supplementary Table 4). In these plants, *FRS7* and *FRS12* loci were re-analysed using TIDE to identify genotypes that were now non-chimeric and either homo- or hetero-allelic (Supplementary Fig. 3c). The T3 progeny CRISPR #3–11, homo-allelic for an out-of-frame mutation at both loci was used for experiments. In this line, both for *FRS7* and *FRS12* the mutation resulted in loss of a restriction enzyme recognition site (Supplementary Fig. 3d). This allows easy genotyping of the mutants using a Cleaved Amplified Polymorphic Sequences (CAPS) assay (Supplementary Fig. 3e,f and Supplementary Table 4).

The *Pro35S:FRS7-GFP* and *Pro35S:FRS12-GFP* constructs were made by Gateway cloning (Invitrogen) using the pFAST-R05 vector to transform Col-0 wt plants[50]. The initial pDONR223-FRS7 and pDONR223-HON4 vectors were obtained from the ABRC plasmid stock (https://abrc.osu.edu/), whereas *FRS12* was PCR-amplified from cDNA and subsequently introduced into pDONR207 (see Supplementary Tables 4 and 5 for the primers and clones used). Promoters of *FRS7*, *FRS12* and *PIF4* were PCR-amplified from Col-0 genomic DNA and subsequently introduced into pENL4R1 or pDONR207 vectors. The *GI* promoter was kindly provided by Prof G. Coupland. *ProFRS7:fLUC* and *ProFRS12:fLUC* constructs were cloned through Multisite Gateway (Invitrogen) cloning into pm42W7 destination vector. *ProPIF4-min35S:fLUC* and *ProGI:fLUC* were cloned through Gateway cloning into pGWLuc destination vector that was kindly provided by Prof G. Coupland. *ProFRS7:GUS* and *ProFRS12:GUS* constructs were made by Gateway cloning the respective promoters into the pmK7S*NFm14GW destination vector[51]. For tandem affinity purification and tandem chromatin affinity purification, FRS12 was carboxy-terminally fused to the GS of HBH tags using the pKCTAP vector[28,52]. For BiFC the FRS7, FRS12 and HON4 were fused amino-terminally to the nGFP of cGFP tags using the pH7m24GW2 or pKm24GW2 vectors, respectively. For tobacco trans-activation assays FRS7 and FRS12 were Gateway cloned into the pGAL4DB vector. The *ProFRS7:FRS7-HA*, *ProFRS12:FRS12-HA*, *Pro35S:FRS12-GR*, *Pro35S:GFP-GR*, *Pro35S:FRS7-HA* and *Pro35S:FRS12-HA* constructs were made through Multisite Gateway cloning into pK7m34GW or pK7m34GW-FAST[53].

**Growth conditions and measurements.** Hypocotyl elongation assays were carried out as described[54] with some modifications. Briefly, seeds were surface-sterilized and sown in ½ Murashige & Skoog (MS) medium with 0.8% agar without sucrose. Seeds were stratified at 4 °C in CD for 3 days, and placed horizontally in SD or LD growth for 10 days with a light intensity of 80 μmol m$^{-2}$ s$^{-1}$. Hypocotyls were extended in new MS plates, plates were then scanned and the hypocotyl lengths were quantified using ImageJ 1.46r software (http://imagej.nih.gpv/ij/). All values represent the mean of 16 to 20 seedlings, significant differences between genotypes and Col-0 wt were assessed using *t* test. Rosette leaf growth and movement dynamics were followed using a high-resolution *in vitro* growth imaging system (IGIS)[23]. *Arabidopsis* plants ($n = 25$) were grown randomly in petri dishes mounted on a rotating disk for 21 days, and images of each plate were taken in an hourly basis. Automated image analysis was applied to measure the rosette area for 21 days in LD growth. The experiment was repeated three times with similar results; sample size was chosen according to pre-established methods[23]. For leaf series analysis, plants ($n = 8$ per genotype) were grown in LD for 18 or SD for 30 days. After dissection of individual leaves, the leaf area was measured with ImageJ (http://rsb.info.nih.gov/ij/) Statistically significant area differences between genotypes and Col-0 wt were assessed using *t* test (Supplementary Table 1). For flowering time assays, seeds were surface-sterilized and sown in ½ MS-agar medium, stratified and germinated at 21 °C for 5 days under LD (16-h light/8-h dark) or SD (8-h light/16-h dark). After this period, seedlings were transferred randomly to soil and grown under the same photoperiodic and temperature conditions. The number of visible rosette and cauline leaves was recorded when the first flower bud opened. A total of 12 biological replicates were considered for each line, significant differences between genotypes and Col-0 wt were assessed using *t* test. Experiment was repeated three times with similar results; sample size was chosen according to pre-established methods[16].

**Subcellular protein localization.** GFP was monitored in primary root cell tips of 5-day-old plants grown vertically on MS medium. Root tips were mounted on slides and GFP fluorescence was followed with a Zeiss LSM 710 confocal microscope using × 25 magnification. Experiment was repeated twice with similar results.

**Tandem affinity purification.** TAP experiments were performed on *Arabidopsis* cell cultures (PSB-D) expressing protein G and Streptavidin-binding peptide (GS)-tagged bait. Cell cultures were grown either in continuous darkness or entrained in LD (16:8) conditions for 2 weeks before collecting at ZT4 or ZT20. Protein interactors were identified by mass spectrometry using an Orbitrap mass spectrometer[28]. Proteins with at least two matched high confident peptides, of which at least one is unique, were retained (Supplementary Table 2). Afterwards, nonspecific interactors were filtered out based on frequency of occurrence of the co-purified proteins in a large dataset containing 543 TAP experiments using 115 different baits[27]. Two biological replicates were analysed for each experiment.

**Phylogenetic analysis.** The first 200 amino acids of FHY3 containing the DNA-binding domain were aligned by ClustalW2 (http://www.clustal.org/) to all members of the FRS family to retrieve a conserved region amongst these proteins. Because FRS9 did not present any conservation of this region, it was not included in the analysis. A phylogenetic tree was subsequently constructed by the Maximum-likelihood method applying 1,000 bootstrap replicates using MEGA5 (ref. 55).

**GUS histochemical analysis.** Histochemical GUS staining was performed in 14 day-old homozygous seedlings expressing *ProFRS7:GUS* and *ProFRS12:GUS* germinated under SD conditions. The plant material was incubated at 37 °C, in the dark, for two hours in a staining buffer containing 1 mM 5-bromo-4-chloro-3-indolyl β-D-glucopyranoside sodium salt (X-Gluc), 0.5% Triton X-100, 1 mM 5-ethylenediaminotetraacetic acid (EDTA) pH 8, 0.5 mM potassium ferricyanide ($K_3Fe(CN)_6$), 0.5 potassium ferricyanide ($K_4Fe(CN)_6$) and 500 mM sodium phosphate buffer pH 7. The reaction was terminated by replacing the staining buffer with 70% ethanol. The material was mounted in 25% lactic acid; 50% glycerol and analysed with a Nikon AZ100M stereo microscope equipped with an AZ Plan Fluor × 5 objective. Experiment was repeated twice with similar results.

**Luminescence and circadian rhythm analysis.** Luminescence analysis was carried out in protoplasts isolated from 4-week-old *Arabidopsis* Col-0 plants grown under a 12L:12D cycle[56]. *ProFRS7:fLUC* of *ProFRS12:fLUC* reporter plasmids were prepared for transfection by CsCl gradient purification and the DNA concentration was adjusted to 3 µg µl$^{-1}$. Overall, 200 µl of protoplast cells were transfected with 5 µl of the reporter plasmids through PEG-mediated transfection. For luminescence analysis, transfected protoplasts were incubated under constant red light condition (30 µmol m$^{-2}$ s$^{-1}$) at 22 °C. Data were imported into the Biological Rhythms Analysis Software System and analysed by FFT-NLLS[57].

**Immunoblotting.** Total protein was extracted using extraction buffer (25 mM Tris/HCl pH 7.6, 15 mM $MgCl_2$, 150 mM NaCl, 15 mM *p*-nitrophenyl phosphate, 60 mM β-glycerophosphate, 0.1% NP-40, 0.1 mM $Na_3VO_4$, 1 mM NaF, 1 mM PMSF, 1 µM E64, complete proteinase inhibitor (Roche), 5% ethylene glycol) and the protein concentration was determined using Bradford assay (Bio-Rad). Samples were denatured in Laemmli buffer, run on a 4–15% TGX gel (Bio-Rad) for 20 min at 300 V, and subsequently blotted on a 0.2 µm PVDF membrane (Bio-Rad). Antibodies used were anti-HA (1:1,000 dilution, 3F10, Roche) and anti-actin8 (1:2,000 dilution, A0480, Sigma). Chemiluminescent detection was performed with Western Bright ECL (Isogen, http://www.isogen-lifescience.com/).

**Bimolecular fluorescence complementation.** Split-GFP (nGFP and cGFP) constructs were transiently expressed by *A. tumefaciens*-mediated transformation of lower epidermal leaf cells of 3- to 4-week-old *N. benthamiana* plants using an infiltration buffer composed of 10 mM $MgCl_2$, 10 mM MES and 100 µM acet-osyringone, and addition of a P19-expressing *Agrobacterium* strain to boost protein expression. All *Agrobacterium* strains were grown for 2 days, diluted to OD 1 in infiltration buffer and incubated for 2 h at room temperature before mixing in a 1:1 ratio with other strains and injecting. Three days after injection, interaction of the proteins was scored by screening lower epidermal cells for fluorescence using a Zeiss LSM 710 confocal microscope at × 25 magnification. Experiment was repeated twice with similar results.

**Transactivation assays.** Transient expression assays (TEA) were carried out in *N. tabacum* (tobacco) protoplasts and *N. benthamiana* leaves[39,58]. Protoplasts were prepared from a Bright Yellow-2 tobacco cell culture and co-transfected with a reporter plasmid containing the fLUC reporter gene and the effector. For each experiment, 2 µg of each plasmid was used and total effector amount was equalized in each experiment with a mock effector plasmid. After transfection, protoplasts were incubated overnight and then lysed. fLUC and rLUC activities were determined with the Dual-Luciferase reporter assay system (Promega). Variations in transfection efficiency and technical error were corrected by normalization of fLUC by rLUC activities.

Leaves of *N. benthamiana* plants were co-infiltrated with *A. tumefaciens* cultures containing the effector and reporter constructs. Three days after inoculation, 0.5 cm diameter leaf discs were collected from the leaves and transferred, with the abaxial side upwards, to 96 well microtiter plates filled with 165 µl ½ MS liquid media and 35 µl of 1 × D-Luciferin substrate (20 µg/µl). A minimal *CAMV35S* promoter region ( − 46 to − 1) was fused at the 3′ end the *ProPIF4*, to enhance endogenous activation of the *fLUC* reporter.

In all cases, eight technical repeats were considered and significant differences were assessed using *t* test. Luciferase activity was measured with BMG LUMIstar Galaxy.

**Tandem chromatin affinity purification–Seq.** Tandem chromatin affinity purification (TChAP)–Seq was performed on 7-day-old *Pro35S:FRS12-HBH-* and *Pro35S:NLS-GFP-HBH*-expressing PSB-D cell cultures transferred to LD conditions, two weeks before collecting at night-time at ZT20. Chromatin was isolated

from formaldehyde-treated cell cultures following two affinity purification steps; first by IMAC using a Ni-NTA Superflow resin (Qiagen), then by a Biotin binding step using a Streptavidin Sepharose resin (GE Healthcare). Finally, protein–DNA bound fragments were decrosslinked, deproteinized and purified using the QIAquick PCR Purification Kit (Qiagen)[52]. Experiments were carried in two replicates. The TChAP DNA samples were processed by first preparing a Trueseq ChIP-seq library (Illumina) and then sequencing using Illumina HiSeq 2000 at 50 bp single read at an average depth of 15 million reads at GATC biotech Ltd, Germany. The quality of the raw data was evaluated with FASTQC (v0.10.0; http://www.bioinformatics.bbsrc.ac.uk/projects/fastqc/), and adaptors and other overrepresented sequences were removed using the fastx-toolkit (v0.0.13; http://hannonlab.cshl.edu/fastx_toolkit/). Reads mapping and filtering to the unmasked TAIR10 reference genome of *Arabidopsis* using BWA with default settings for all parameters (TAIR10_chr_all.fas; ftp.arabidopsis.org)[36]. Peak calling was performed using MACS v2.0.10 (ref. 59) at default parameters except − g 1.0e8 and FDR < 0.05, adding a cutoff stringency of 10. Peak regions were annotated based on the location of their summits. A peak was assigned to the closest gene as annotated in the TAIR10 release represented in the PLAZA2.5 database[60]; peaks can be assigned both 5′ and 3′ of a gene. Each assignment was considered as a potential TF-target interaction. The peak locations were categorized by assigning a peak to one of the following genomic regions: intergenic, 1 kb promoter (1 kb upstream of Transcription Start Site), 5′UTR, coding, intron, 3′UTR, and 1 kb down of the transcription stop site. *De novo* motif finding was carried using peak motifs[37]. The *P* value for motif enrichment in the peak set compared with the genomic background was calculated empirically. For the TF co-binding matrix, the TFs were clustered based on the Jaccard distance (1 − Jaccard Index) between their target sets using average linkage hierarchical clustering[36]. The overlap was validated statistically using the hypergeometric *P* value, with Bonferroni correction for multiple hypothesis testing. The cutoff for significance was set at 0.001 (Supplementary Fig. 8c). Information about total read counts, filtered and mapped reads is presented in Supplementary Table 6. To integrate the FRS12 TChAP-Seq and DE genes with bound genes of PIF4 and the circadian clock, ChIP-Seq data from PIF4, CCA1, TOC1, PRR3, PRR5 and PRR7 were obtained from previously published datasets[29,61,62]. Selected TF-target gene pairs were visualized using Cytoscape 3.2.0.

**Chromatin immunoprecipitation–qPCR analysis.** Seedlings expressing *ProFRS12:FRS12-HA* and Col-0 wt were grown in MS-agar medium for 10 days, harvested in liquid nitrogen at the indicated time points and stored at − 80 °C before analysis. Plant material was grinded, lysed and crosslinked using formaldehyde. The crosslinking reaction was stopped using glycine. Nuclei were isolated and lysed in a sucrose gradient and the chromatin obtained was fragmented by sonication (Bioruptor Next Gen, Diagenode). Immunoprecipitations were performed using anti-HA (3F10 Roche)-coated IgG magnetic beads (Dynabeads protein G 1003D, Invitrogen). Protein–DNA complexes were eluted using 0.1M $NaHCO_3$ and 1% w/w SDS. Reverse DNA crosslinking was performed in two steps: overnight at 65 °C using 0.2 M NaCl and 1 h incubation at 45 °C adding 10 µl 0.5 M EDTA, 20 µl 1 M Tris HCl pH 6.5 and 2 µl 10 mg ml$^{-1}$ proteinase K. DNA was extracted using phenol-chloroform isoamyl alcohol pH8 and precipitated with sodium acetate/glycogen/ethanol[63]. The qPCR analysis was performed with 0.5 µl of sample per reaction using and the LC480 SYBR Green I Master Kit (Roche Diagnostics) on a Roche LightCycler 480 system. For each pair of primers, normalization to DNA input was carried out. The fold enrichment was calculated as relative to Col-0 wt using the ΔCt expression values.

**RNA isolation and qPCR analysis.** RNA was extracted using the RNeasy Plant Mini Kit (Qiagen), and treated with DNase I (Promega) prior to complementary DNA (cDNA) synthesis with the iScript cDNA Synthesis Kit (Bio-Rad) according to the manufacturer's instructions. Relative transcript abundance of selected genes (for a list of genes and the primers used, see Supplementary Table 4) was determined using the Roche LightCycler 480 system and the LC480 SYBR Green I Master Kit (Roche Diagnostics). Measurements were taken for three technical repeats. The amplification data were analysed using the second derivative maximum method, and resulting cycle threshold values were converted into relative expression values using the comparative cycle threshold method.

**RNA-Seq analysis.** Stratified seedlings of *Pro35S:FRS12-GR* and *Pro35S:GFP-GR* were grown in ½ MS with 1% sucrose liquid medium under LD conditions and treated with mock or 5 µM dexamethasone 4 h before harvesting at night (ZT20). Three biological replicates were harvested for each line. RNA was isolated using the RNeasy Plant Mini Kit (Qiagen) and DNase I treated (Promega). RNA samples were processed by first preparing a Trueseq RNA-Seq library (Illumina) and then sequenced at 30 million reads at 50 bp single read using Illumina HiSeq 2000 technology at GATC Biotech, Germany. Read quality control, filtering, mapping to the TAIR10 *Arabidopsis* genome and read counting were carried out using the Galaxy portal running on an internal server (http://galaxyproject.org/). Sequences were filtered and trimmed, respectively, with the Filter FASTQ v1 and FASTQ Quality Trimmer v1 tools with default settings (http://www.bioinformatics.babra-ham.ac.uk/projects/fastqc/). Reads were subsequently mapped to the TAIR10

version of the *Arabidopsis* genome using GSNAPv2 allowing a maximum of five mismatches. The concordantly paired reads that uniquely mapped to the genome were used for quantification on the gene level with htseq-count from the HTSeq python package. Data was normalized using TMM and common dispersion was then estimated using the conditional maximum-likelihood method implemented in edgeR[64]. Differentially expressed genes were defined by a 2-fold difference between samples with corrected *P* value <0.05 at a FDR <0.05. GO-enrichment analysis was carried with Cytoscape 2.8.2 software using the BINGO plugin[65] in default mode running with actualized GO and gene annotation definitions for *A. thaliana* from 26 January, 2016 (http://geneontology.org/).

**Data availability.** The RNA-Seq and TChAP-Seq datasets are available at the EMBL-EBI database under accession codes E-MTAB-3018 and E-MTAB-3019, respectively. The authors declare that all other data supporting the findings of this study are available within the manuscript and its supplementary files or are available from the corresponding author upon request.

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

## Acknowledgements

We thank Annick Bleys and Blanche Magarinos-Rey for help in preparing the manuscript and Veronique Storme and Frederik Coppens for their assistance in the statistical and bioinformatics analyses. We thank Dr Salomé Prat for providing the *Agrobacterium tumefaciens* strains containing the *ProPIL1:fLUC* and *35S:PIF4-HA* constructs; and Prof George Coupland for providing the *ProGI*-pDONR207 and the pGWLuc vectors. This study was funded by the Belgian Science Policy organization for a postdoctoral fellowship to A.R. and S.I., by the Research Foundation Flanders for a postdoctoral fellowship to P.F.-C., L.P. and S.D., and for project G005312N and G005915N, by the EMBO for a short-term fellowship to P.F.-C. and by the Marie Curie Actions programme FP7-PEOPLE-2013-IEF for a postdoctoral fellowship to P.F.-C. K.V. acknowledges the Multidisciplinary Research Partnership 'Bioinformatics: from nucleotides to networks' Project (No. 01MR0310W) of Ghent University. D.E.S. was supported by the National Institutes of Health Grant R01GM093285.

## Author contributions

A.R., S.I., P.F.-C., L.P. and A.G. designed the research. A.R., S.I., P.F.-C., H.S., L.D.M., R.V.B., R.D.C., D.E., M.R. and L.P. performed experiments. A.R., K.S.H. and S.D. performed bioinformatics analysis. A.R. and A.G. wrote the article. A.G., L.P., D.E.S., P.F.-C., D.I., K.V., K.S.H., D.E., D.E.S., K.G. and G.D.J. interpreted and commented the article.

## Additional information

**Competing interests:** The authors declare no competing financial interests.

**Publisher's note**: 

**DOI: 10.1038/ncomms16213**

# Author Correction: The transcriptional repressor complex FRS7-FRS12 regulates flowering time and growth in *Arabidopsis*

Andrés Ritter, Sabrina Iñigo, Patricia Fernández-Calvo, Ken S. Heyndrickx, Stijn Dhondt, Hua Shi, Liesbeth De Milde, Robin Vanden Bossche, Rebecca De Clercq, Dominique Eeckhout, Mily Ron, David E. Somers, Dirk Inzé, Kris Gevaert, Geert De Jaeger, Klaas Vandepoele, Laurens Pauwels & Alain Goossens

*Nature Communications* 8:15235 doi: 10.1038/ncomms15235 (2017); Published online 11 May 2017; Updated 10 Apr 2018

The financial support for this Article was not correctly acknowledged. The Acknowledgements incorrectly stated that the study was funded in part by Research Foundation Flanders for project GN005212.

The Acknowledgements should have stated that the study was funded in part by Research Foundation Flanders for projects G005312N and G005915N.

This error has been corrected in both the PDF and the HTML versions of the Article.

