## [Peer Review File · Nature Communications]

Reviewers' comments:

Reviewer #1 (Remarks to the Author):

The manuscript by Ritter et al. describes the identification of FRS7 and FRS12 as transcription factors that negatively regulate genes involved in promotion of flowering and elongation growth in the model plant, *Arabidopsis*, in a photoperiod-dependent manner.

FRS7 and 12 proteins accumulate maximally in short days (SD) with a peak around dusk. *frs7* and *frs12* mutants show excessive hypocotyl elongation specifically in SD, while over-expressors show short hypocotyls. Similarly, *frs7* and *frs12* mutants show greater rosette expansion, while over-expressors show reduced rosette expansion (although these data are only presented for LD - see comment below). *frs7 frs12* double mutants also show early flowering, more dramatically in SD.

FRS7 and 12 localise in the meristem and vasculature, they bind to each other in vivo and form a nuclear-localised complex. Both are shown to be able to repress gene expression when bound to a gene promoter. Inducible FRS12 expression was used to identify genes whose expression is regulated by FRS12. Genes repressed by FRS 12 possess circadian clock, photoperiodic processes and red-light signalling among the most significantly enriched terms

Directly-regulated target genes of FRS12 are identified based on a combination of TChAPseq and RNA seq and these genes include a number of genes related to flowering time and diurnal growth pathways which are negatively regulated by FRS12. Transient expression assays confirm the trans-repression of the promoters of three of these key growth / flowering genes, PIF4, PIL1 and GI by both FRS7 and FRS12.

Three potential FRS12 binding sites are identified by enrichment analysis among target genes. In one case, GI, the binding site overlaps the evening element responsible for clock regulation of GI expression suggesting that FRS12 may physically interfere with binding of circadian transcription factors. The biggest effect of the evening-expressed FRS12 in such a case would be prevention of binding of positively acting factors promoting GI expression at that time. Consistent with this, GI expression shows a higher evening peak in SD in the *frs7 frs12* double mutant. The *frs7 frs12* double mutant also shows increased amplitude of several other light/clock regulated genes involved in positive regulation of elongation growth or transition to flowering. These patterns are entirely consistent with the elongated hypocotyls and early flowering of *frs7 frs12* double mutants in these short day conditions.

Overall, this is a convincing and impressive body of work. It reveals a completely new mechanism of regulation in plants which is clearly an important part of appropriate adaptation to photoperiod. The work has important implications across a wide range of plant science research in addressing both growth and flowering and these photoperiodic responses are an environmental adaptation that has important agricultural implications. Both the quantity and quality of the data is impressive. The methodology is entirely valid and sound scientific method is applied to reveal a complete pathway. The findings are internally consistent and all appropriate controls are included. The presentation of the work is also of very high quality. Abstract, introduction and conclusion are all appropriate and the resultant manuscript is clear, easy to follow and accessible to a wide audience.

Three loose ends should be tidied up with simple experiments.

FRS12 gene transcription peaks during the subjective day in constant red light and a degradation assay shows that FRS12 is degraded in light. However, these data are not consistent with the peak of FRS12 at the 8 h time point in SD. Assuming the 8h time point is taken exactly at dusk in SD then there would be no time for FRS12 protein to accumulate following lights-off. If FRS12 gene transcription peaks at the same time in SD as in constant red light, transcription at that point is actually lower than earlier in the day so this cannot account for the FRS12 protein peak at 8h in SD. I think it is additionally necessary to look at FRS12 transcription / RNA levels in a diurnal cycle (particularly SD) as the protein peak suggests that this may not match the pattern of transcription in constant light.

Rosette / leaf expansion should be analysed in SD. Given the protein expression pattern and the effects on hypocotyl elongation and flowering, it would be expected that this phenotype would also be more dramatic in SD. It is necessary to determine whether this is also a "photoperiodic growth" phenotype as per the title and abstract.

frs7 frs12 double mutants show increased evening peaks of expression of direct targets GI, PIF4 and CO in SD. For completeness, gene expression should also be added for LD. According to the authors' conclusions based on the protein expression pattern, hypocotyl growth and flowering time data from the double mutant, it would be expected that there should be a much lesser effect in LD. It would be easy to test this.

In addition, I suggest additional replicates of the following specific experiments in supplementary figure 6 should be added to tighten these data. GI data for the double mutant in supplementary figure 6 does not replicate fig 6a. Although the difference is qualitatively the same, the large error bars in supplementary fig 6 mean that the difference does not show as significant here. Similarly, PIL1 shows a difference at ZT8 in the double mutant in supplementary figure 6 but no difference in the double mutant at this "ZT8" time point in figure 6a. Again large error bars are apparent for the data in supplementary figure 6.

Minor points:

Page 9, line 8 should read "potentially represses large portions of PIF4 and *AP1* downstream gene targets".

Page 9, line 21: "The FRB1 was found significantly enriched in peak summits of the PIF4 and GI promoters whereas FRB2 and FRB3 were found enriched in peaks of the PIL1 promoter". As I understand it the FRB sequences found in the peak summits are single occurrences so it is not correct to describe this as "enrichment".

Page 11, line 12: "Enhanced expression of these genes was mostly observed only in the frs7-1;frs12-1 double mutant but not in the single frs7-1 or frs12-1 mutants." This is not strictly true. In 50% of the cases a significant effect can be seen for the single frs7 mutant meaning that this sentence needs rephrasing.

The figure legend to supplementary figure 5 refers to coloured arrows and rectangles. However, in the figure itself these appear to be black.

In figure 5 b the lowest panel shows sequence from the GI promoter. Some blocks of two or four bases are coloured yellow or blue. What does this represent?

qPCR results in figure 6 record "relative expression". What does "1" on the scale represent?

Reviewer #2 (Remarks to the Author):

The authors found that two FRS family members, FRS7 and FRS12, interact and form a protein complex termed "Saturnalia Complex (SC)" that regulates hypocotyl elongation and flowering time, especially predominantly in short days. This is a novel interaction that is put into mechanistic context with a complementary mix of genetics and biochemistry. RNA-seq and tandem chromatin affinity purification coupled with next generation sequencing (TChAP-seq) identify target genes of the SC. Chromatin immunoprecipitation (ChIP) assays and gene expression analysis support PIF4 and GI genes are direct targets of the SC for short day-dependent hypocotyl elongation and flowering time control, respectively. The manuscript proposes a model for the daylength-dependent formation of repression complex that the active SC only exists in short days but not in long days. This is a meticulous and somehow intriguing piece of research focused on photoperiodic regulation of growth and development in Arabidopsis. However, the data by itself doesn't support the conclusions well. In general, the text is logical but not well written with many awkward sentences. The authors often tend to conclude their findings without further supportive data.

Major points;

1. The authors claim that FRS12 protein is degraded by light in a proteasome-dependent manner based on supplementary figure 1b. However, as shown in figure 1b, the abundance of FRS12 protein appears to peak at dusk in short days, and the protein is stabilized the most under the light period in long days as well.

2. FRS7 and FRS12 proteins interact in vivo but the authors didn't provide when formation of the SC occurs. Day or night?

Figure 6b illustrates that longer night condition allows the proteins to form the active SC, which can repress target genes, and that, by contrast, shorter night condition doesn't. As shown in figure 1b, drastic changes in protein abundance between short and long days are obvious but it looks that substantial amounts of the two proteins still remain in long days, indicating that the active SC might exist even in this conditions. It would be good to provide the photoperiod-dependent interaction.

3. Concerns above raise an important question that if there is a dosage-dependent repression by the SC. If it's the case, we will see more drastic changes in target gene expression under short days than in long days. To compare and quantify the relative expression levels under different photoperiod, qRT-PCR needs to be performed using samples from frs7/12 double mutant as well as in at least two independent FRS7 or FRS12 overexpression lines such as Pro35S:FRS7-HA or Pro35S:FRS12-HA with a 24-h rhythm, a whole day time course, rather than a single time point shown in supplementary figure 4.

4. Quantitative ChIP assays are needed to support daylength-dependent the association of SC with PIF4 and GI promoters using ProFRS7:FRS7-HA or ProFRS12:FRS12-HA transgenic plants individually harvested during the light and dark periods and grown in long day and short day conditions.

5. Lastly, the major overarching question raised by based on figure 6a that the authors may misinterpret is:

What is the biological relevance of SC for photoperiod-dependent hypocotyl elongation and flowering time regulated by the SC?

The highest accumulation of PIF4 protein in short days occurs at dawn that enables hypocotyl to be elongated at the end of the night. As the authors are claiming, If the SC represses PIF4 gene expression at night, we would expect increased nighttime PIF4 transcript levels in frs7/12 double mutant. However, shown in figure 6a, only daytime PIF4 expression is slightly increased in the double mutant background.

In addition, unlike PIF4, the role of GI in flowering regulation under short days is restricted. Mutation or overexpression of GI causes a subtle change in flowering under short days, because the expression of FKF1 and GI protein is out of phase that leads to accumulation of floral repressors, CDF proteins that delay flowering time, as the authors mentioned in the manuscript. Therefore, changed GI amount (= elevated GI transcripts) at dusk in short days by the frs7/12 mutations might not correlate with the flowering phenotype of frs7/12 double mutant.

Minor points;

1. For all western blot data, data quality needs to be improved. For example, figure 1b and S1b need to be quantified using an internal control antibody such as -Actin instead of Rubisco staining on membranes.

2. To show FRS genes are the clock-regulated, the authors entrained Arabidopsis protoplasts in 12L/12D and transferred continuous red light, which is unusual condition, and used transient luciferase report system. Any idea why the period is lengthened in the subsequent day in the condition?

Also, are there any reasons not to use continuous white light or dark? Or any specific reasons to use only red light?

I recommend that entrain wild type plants, transferred to either continuous white light or dark and harvest at indicated age and time points. And then run qRT-PCR.

3. In the manuscript, Arabidopsis protoplasts and Arabidopsis cell culture were used. I wonder if those are appropriate materials for the experiments.

Reviewer #3 (Remarks to the Author):

The authors have characterized FRS7 and FRS12, which are circadian and photoperiodic regulated proteins, in this study. These proteins highly accumulate in short-day conditions and have a role in transcription factor. They have demonstrated that these proteins repress hypocotyl growth and regulate flowering time in a photoperiodic-dependent manner using mutant lines and overexpression lines. They have also found some interacting proteins with FRS12 by IP-MS analysis using TAP-tag purification method expressed in cultured Arabidopsis cells. It was confirmed that these candidate proteins were localized with FRS12 in nucleus by BiFC assay. Then, they have found that FRS12 bind to cis-elements on promoter region of some downregulated genes in FRS12 overexpression lines. Please see following comments for improve manuscript.

1. They have showed only 4 identified proteins by IP-MS analysis in Figure 4a and Supplementary Data 1. I would like to make sure that other proteins were not identified or not. Because more proteins including false-positive proteins are usually identified in proteomic analysis using LTQ-Orbitrap Velos. If they choose only these 4 proteins, please show the criteria.
2. I recommend that they should make a new Table and show if unique peptides were assigned to both FRS7 and FRS12. Otherwise, it cannot exclude that only conserved peptides between FRS7 and FRS12 were identified. They should also show unique peptides were assigned to AHL9 and AHL14.
3. P7. L12-14: I could not understand the meaning of "Saturnalia Complex (SC)". Does this name make sense based on their function? If I correctly understand of this meaning, it is not suit to their function.
4. Concerning RNA-seq results, they should describe up-regulated genes bit more. I feel that they just describe down-regulated genes because these genes are convenience to explain their model.
5. P25 (TAP method): They describe the TAP methods very simply in this manuscript. They just cite ref. 26 as method. I would like to ask them if their methods concerning IP-MS are exactly same with ref. 26. How did they prepare Arabidopsis cell cultures with FRS12-TAP? Have they performed in-gel digestion? If so, how many gels were excised? Have they used exactly same HPLC, buffer, column, method and so on? If they modify concerning some steps, they should describe details. Otherwise, readers cannot reproduce their experiments.
6. Please explain briefly concerning min35S:fLUC method. I could not understand correctly this method.
7. P6, Line 8, Co-0 should be Col-0.

Reviewer #4 (Remarks to the Author):

The authors proposed that FRS7 and FRS12 proteins are involved in seasonal growth and

flowering time. To conclude this, they performed quite broad experiments, most of which were with cutting edge methods (gene expression analysis, ChIPseq analysis, gene reporter assay, and phenotypes observation). This is first time to report that these FRS genes are implicated in seasonal responses in plants.

The paper is well written and easy to follow, and may have impact on related research fields.

However, I have some concerns about data in the manuscript.

1. Although authors described that *frs7 frs12* double mutants and over-expression lines of FRSs showed opposite phenotypes, but they used only one allele of *frs7* and *frs12*, and any complementation tests were not performed. Phenotypes of mutants are likely, but other genetic loci are possibly responsible for the phenotypes.

2. They described that expression of FRS7 and FRS12 are under the clock control, from the luminescence reporter assay using mesophyll protoplasts. However, public database (DIURNAL, <http://diurnal.mocklerlab.org/>) show constant mRNA amounts of these genes under diurnal and circadian conditions. In addition, amplitude of luminescence rhythms are relatively lower, compared with those driven by canonical clock genes promoters. FRS12 promoter activity is mainly in leaf vasculature, not in mesophyll (Figure 3a). I feel that more careful description are required for expression of these FRSs.

3. They performed ChIP-seq (TChAp-seq) using cultured cell, but it is not clear whether FRS12-HBH fusion protein expressed in these cells are biologically active (native FRS12 is expressed in vasculature). This may call claim the result of TChAp-seq (Do binding sites are reflected of biological function of FRS12?). Information about reads (or depth) of mapping data (Figure 5) are not shown.

4. Related to TChAp-seq. The manuscript contains comparison data among previous ChIPseq studies and FRS12 ChIPseq, but lacks the accurate knowledge for gene regulatory network under the clock (ChIPseq study of CCA1, TOC1, and PRRs; <http://www.ncbi.nlm.nih.gov/pubmed/?term=chip%2C+circadian%2C+ara+bidopsis>). PIF4 gene is bound by TOC1 and PRRs, and GI gene is bound by CCA1. Comparison of previous studies and this manuscript may provide further insight for gene regulatory network for photoperiodic flowering time and growth control.

5. They suggest that regulation of GI expression by SC is involved in photoperiodic flowering pathway. In this scenario, expression of FKF1, a critical partner of GI for flowering induction, is also important. However, FKF1 expression in *frs* mutants are not shown.

6. Expression of PIF genes for light and clock dependent growth is mainly in epidermis (reviewed in, <http://www.ncbi.nlm.nih.gov/pubmed/26723003>), and this is important for growth. However, FRS expression are in vascular, which may not control PIF expression directly in epidermis.

7. They suggest that FRS proteins function in dark period, but expression changes of GI and PIF4 (FRS-targets) are in daytime (Figure 6). How do they explain the difference of

accumulation time of FRS and effecting time on FRS-targets.

Minor comments.

1. Figure 4e, fold change is ">" 2.
2. page9, line10, GI downstream may be "AP1" downstream.
3. PRR7 expression is downregulated by FRS12-GR (Figure 4), but it is not changed in frs7 frs12 (Figure 6). What do the data tell us?

Response to the reviewers' comments:

Reviewer #1:

Overall, this is a convincing and impressive body of work. It reveals a completely new mechanism of regulation in plants which is clearly an important part of appropriate adaptation to photoperiod. The work has important implications across a wide range of plant science research in addressing both growth and flowering and these photoperiodic responses are an environmental adaptation that has important agricultural implications. Both the quantity and quality of the data is impressive. The methodology is entirely valid and sound scientific method is applied to reveal a complete pathway. The findings are internally consistent and all appropriate controls are included. The presentation of the work is also of very high quality. Abstract, introduction and conclusion are all appropriate and the resultant manuscript is clear, easy to follow and accessible to a wide audience. Three loose ends should be tidied up with simple experiments.

1. FRS12 gene transcription peaks during the subjective day in constant red light and a degradation assay shows that FRS12 is degraded in light. However, these data are not consistent with the peak of FRS12 at the 8 h time point in SD. Assuming the 8h time point is taken exactly at dusk in SD then there would be no time for FRS12 protein to accumulate following lights-off. If FRS12 gene transcription peaks at the same time in SD as in constant red light, transcription at that point is actually lower than earlier in the day so this cannot account for the FRS12 protein peak at 8h in SD. I think it is additionally necessary to look at FRS12 transcription / RNA levels in a diurnal cycle (particularly SD) as the protein peak suggests that this may not match the pattern of transcription in constant light.

We thank the reviewer for pointing to these inconsistencies. As suggested, we have therefore looked in more detail at *FRS7* and *FRS12* transcript levels in the diurnal cycle in LD and SD. Through qPCR analysis we now show that Arabidopsis wild type plants display higher expression of *FRS7* and *FRS12* throughout the diurnal cycle in SD as compared to LD growth (new Figure 1b). This observation is in accordance with our results obtained at the protein level (Figure 1d).

In SD, both *FRS7* and *FRS12* gene expression peak at dawn. This phase of expression is in agreement with the transcriptional activity of the *FRS7* and *FRS12* promoters as shown in Figure 1a. These results demonstrate that *FRS7* and *FRS12* are particularly transcribed in SD at dawn.

Notably however, diurnal expression patterns in SD of *FRS7* and *FRS12* proteins show a shift in expression, accumulating during daytime, peaking at dusk, and then decreasing during night time (Figure 1c). In accordance with these results we have now adapted the corresponding results (page 4, lines 84-98) and discussion sections.

Finally, considering all of these results, we realised that the light stability assay previously shown in Supplementary Figure 1b was too artificial as it did not reflect the conditions that regulate photoperiodic or diurnal expression of *FRS7* and *FRS12* proteins (see also our response to comment#1 from reviewer#2). Therefore it has been removed from the revised manuscript.

2. *Rosette / leaf expansion should be analysed in SD. Given the protein expression pattern and the effects on hypocotyl elongation and flowering, it would be expected that this phenotype would also be more dramatic in SD. It is necessary to determine whether this is also a "photoperiodic growth" phenotype as per the title and abstract.*

The suggested experiment has been carried out. More particularly, areas of rosette leaf series of Col-0 wt, *frs7-1;frs12-1*, *Pro35S:FRS7-HA-1* and *Pro35S:FRS12-HA-1* lines have been measured after 30 days in SD growth (and compared to 18 days in LD). The results are presented in the revised Figure 3, Supplementary Figure 2 and Supplementary Table 1 and described in the results section on page 5-6, lines 118-135. This experiment reproduced the previously obtained results in LD (now partly shown in Supplementary Figure 2) and found a similar trend in SD growth (Figure 3b-c). However, contrary to the expectations of the reviewer, the effect was less dramatic in SD than LD. As plants grow slower in SD [see Cookson et al. *Ann. Bot.* 99, 703-711 (2007)], differences in rosette areas become more obvious only at stages later than 30 days (see the new Figure 3d). However, at later stages other developmental factors, such as flowering and leaf senescence, make accurate comparisons of rosette and leaf areas technically difficult, and, unfortunately, incompatible with our current experimental/technological set-up. Hence, we apologise we were not be able to further assess rosette/leaf expansion for longer time periods in SD.

3. *frs7 frs12 double mutants show increased evening peaks of expression of direct targets GI, PIF4 and CO in SD. For completeness, gene expression should also be added for LD. According to the authors' conclusions based on the protein expression pattern, hypocotyl growth and flowering time data from the double mutant, it would be expected that there should be a much lesser effect in LD. It would be easy to test this.*

We have included expression analysis for the indicated genes, as well as a few others, such as *FKF1* (see also comment #5 of reviewer#4) and the PIF4 targets *PIL1* and *HFR1*, by qPCR of *FRS7/FRS12* overexpressing lines and the double *frs7-1;frs12-1* mutant in both SD and LD conditions (new Figure 7 and Supplementary Figures 9 and 10). Though effects are often visible either at SD or LD photoperiods, for some genes in some lines the effect was more apparent under SD indeed (see results section page 13-14, lines 295-325) (see also comment#3 of reviewer#2.

4. *In addition, I suggest additional replicates of the following specific experiments in supplementary figure 6 should be added to tighten these data. GI data for the double mutant in supplementary figure 6 does not replicate fig 6a. Although the difference is qualitatively the same, the large error bars in supplementary fig 6 mean that the difference does not show as significant here. Similarly, PIL1 shows a difference at ZT8 in the double mutant in supplementary figure 6 but no difference in the double mutant at this "ZT8" time point in figure 6a. Again large error bars are apparent for the data in supplementary figure 6.*

We thank the reviewer for noting these discrepancies in our data. We have repeated all experiments (also to include LD data, see the comment above). Essentially, we consistently observe similar trends between experiments, but the effect in the double *frs7frs12* mutant is only statistically significant for *PIF4* expression (see new Figure 7 and Supplementary Figures 9 and 10).

5. Minor points:

*a. Page 9, line 8 should read "potentially represses large portions of PIF4 and *AP1* downstream gene targets".*

The text has been corrected accordingly.

b. Page 9, line 21: "The FRB1 was found significantly enriched in peak summits of the PIF4 and GI promoters whereas FRB2 and FRB3 were found enriched in peaks of the PIL1 promoter". As I understand it the FRB sequences found in the peak summits are single occurrences so it is not correct to describe this as "enrichment".

The text has been corrected accordingly.

*c. Page 11, line 12: "Enhanced expression of these genes was mostly observed only in the *frs7-1;frs12-1* double mutant but not in the single *frs7-1* or *frs12-1* mutants." This is not strictly true. In 50% of the cases a significant effect can be seen for the single *frs7* mutant meaning that this sentence needs rephrasing.*

The text has been corrected accordingly (see also our reply to comment#4 of the reviewer).

d. The figure legend to supplementary figure 5 refers to coloured arrows and rectangles. However, in the figure itself these appear to be black.

The text has been corrected accordingly.

e. In figure 5 b the lowest panel shows sequence from the GI promoter. Some blocks of two or four bases are coloured yellow or blue. What does this represent?

These coloured blocks were not relevant for the understanding of the figure and have therefore been removed.

f. qPCR results in figure 6 record "relative expression". What does "1" on the scale represent?

"1" represents the highest level of expression for a particular gene. All other expression levels are proportional to this value. This has been indicated in the legend of this figure and all other figures with qPCR data.

Reviewer #2:

The manuscript proposes a model for the daylength-dependent formation of repression complex that the active SC only exists in short days but not in long days. This is a meticulous and somehow intriguing piece of research focused on photoperiodic regulation of growth and development in Arabidopsis. However, the data by itself doesn't support the conclusions well. In general, the text is logical but not well written with many awkward sentences. The authors often tend to conclude their findings without further supportive data.

Major points;

1. The authors claim that FRS12 protein is degraded by light in a proteasome-dependent manner based on supplementary figure 1b. However, as shown in figure 1b, the abundance of FRS12 protein appears to peak at dusk in short days, and the protein is stabilized the most under the light period in long days as well.

We agree with the concern of the reviewer. As also indicated in our response to comment#1 from reviewer#1, we realised from the data from the additional expression analysis demanded by reviewer#1 that the light stability assay previously shown in Supplementary Figure 1b was too artificial as it did not reflect the conditions that regulate photoperiodic or diurnal expression of FRS7 and FRS12 proteins. Therefore it has been removed from the revised manuscript. The data presented in the revised Figure 1 now provide a clear view on FRS7/12 expression and accumulation.

2. FRS7 and FRS12 proteins interact in vivo but the authors didn't provide when formation of the SC occurs. Day or night?

Figure 6b illustrates that longer night condition allows the proteins to form the active SC, which can repress target genes, and that, by contrast, shorter night condition doesn't. As shown in figure 1b, drastic changes in protein abundance between short and long days are obvious but it looks that substantial amounts of the two proteins still remain in long days, indicating that the active SC might exist even in this conditions. It would be good to provide the photoperiod-dependent interaction.

We thank the referee for this suggestion. We agree with the reviewer that substantial amounts of FRS7 and FRS12 proteins are still present in LD. In this context, we do not expect the exclusive assembly of the complex in SD, but rather a dose dependent effect pending on FRS7 and FRS12 protein abundances. To address this comment, we have carried out new tandem affinity purifications with tagged FRS12 proteins as bait, but this time not in Arabidopsis cells growing in continuous dark, but in Arabidopsis cells growing in LD. Cells grown in this condition were harvested at two distinct time points, i.e. during day time at ZT 4 and night time at ZT 20. This analysis revealed that FRS12 recruits FRS7 proteins in all conditions (new supplementary Dataset 1). In addition this analysis further confirmed the interactions with the histone-like protein HON4 and the AT-hook motif nuclear localized protein AHL14, also in LD conditions. Together, these results demonstrate that the assembly of the FRS7-FRS12 complex is not restricted to a

particular photoperiodic or time of the day condition. This has been described in the results section on page 7 lines 153-169.

3. Concerns above raise an important question that if there is a dosage-dependent repression by the SC. If it's the case, we will see more drastic changes in target gene expression under short days than in long days. To compare and quantify the relative expression levels under different photoperiod, qRT-PCR needs to be performed using samples from frs7/12 double mutant as well as in at least two independent FRS7 or FRS12 overexpression lines such as Pro35S:FRS7-HA or Pro35S:FRS12-HA with a 24-h rhythm, a whole day time course, rather than a single time point shown in supplementary figure 4.

The demanded additional gene expression analysis has been carried out (new Figure 7 and Supplementary Figures 9 and 10). Please see also our reply to comment#3 of reviewer#1 for more details.

4. Quantitative ChIP assays are needed to support daylength-dependent the association of SC with PIF4 and GI promoters using ProFRS7:FRS7-HA or ProFRS12:FRS12-HA transgenic plants individually harvested during the light and dark periods and grown in long day and short day conditions.

We thank the referee for this suggestion. The requested experiment was carried out with the ProFRS7:FRS7-HA line for the PIF4 and GI promoters (see the new Supplementary Figure 8). The results indicate a clear time of the day dependent association of FRS7 to the PIF4 and GI promoters in both LD and SD. FRS7 shows increased binding during early daytime at ZT4 and decreased binding at night time ZT20. Notably, these results are in accordance with the circadian protein accumulation patterns of the FRS7 protein. For the interest of the reviewer, we did not compare between LD and SD conditions, however, as experiments were carried out independently, and hence the comparison SD/LD must primarily be considered on a qualitative level in our opinion.

5. Lastly, the major overarching question raised by based on figure 6a that the authors may misinterpret is:

What is the biological relevance of SC for photoperiod-dependent hypocotyl elongation and flowering time regulated by the SC?

The highest accumulation of PIF4 protein in short days occurs at dawn that enables hypocotyl to be elongated at the end of the night. As the authors are claiming, If the SC represses PIF4 gene expression at night, we would expect increased nighttime PIF4 transcript levels in frs7/12 double mutant. However, shown in figure 6a, only daytime PIF4 expression is slightly increased in the double mutant background.

In addition, unlike PIF4, the role of GI in flowering regulation under short days is restricted. Mutation or overexpression of GI causes a subtle change in flowering under short days, because the expression of FKF1 and GI protein is out of phase that leads to accumulation of floral repressors, CDF proteins that delay flowering time, as the authors mentioned in the manuscript. Therefore, changed GI amount (= elevated GI transcripts) at dusk in short days

by the frs7/12 mutations might not correlate with the flowering phenotype of frs7/12 double mutant.

With regard to the biological relevance of the FRS7-FRS12 complex (the name ‘Saturnalia complex’ has been removed, see comment#3 of reviewer#3), the revised version of our manuscript includes new experimental data, obtained through the experiments suggested by all reviewers, which allowed us to refine our view on FRS7-FRS12 function. We now describe the FRS7-FRS12 complex as a new component of the plant photoperiodic system that is preferentially, but not exclusively, active during short-days. We show that this complex accumulates during daytime and operates by repressing two main clock outputs, namely flowering time and growth. In accordance with this and the comment of the reviewer we have now adapted our conclusions in the corresponding results and discussion sections.

With regard to the daytime effect on *PIF4* expression in the double mutant background: FRS7 and FRS12 proteins accumulate during day time, phasing at dusk, repressing *PIF4* and *GI* at this period of the day. This might constitute a mechanism that acts coordinately with phyB and the clock to transcriptionally repress inappropriate growth during day time.

With regard to the comment on the possible contribution of PIF4 to the flowering phenotypes of *frs7-1;frs12-1* double mutant, we agree with the reviewer. This point has now been added in the revised discussion. Nonetheless, the contribution of PIF4 to photoperiodic flowering time is minor compared to its effect at high temperature conditions [see Galvão et al. *Plant J.* 84, 949-962 (2015) and Fernández et al. *Plant J.* 86, 426-440 (2016)].

Regarding *GI*, previous data show that overexpressing this gene in *Arabidopsis* causes dramatic acceleration in flowering time, especially under short days [see Mizoguchi et al. *Plant Cell* 17, 2255-2270 (2005)]. Furthermore, these authors showed that *GI* overexpression causes upregulation of *CO* without affecting *FKF1* expression. Considering these findings, we believe our interpretation about *GI* is still valid.

6. Minor points;

a. For all western blot data, data quality needs to be improved. For example, figure 1b and S1b need to be quantified using an internal control antibody such as -Actin instead of Rubisco staining on membranes.

Rubisco staining has been used as loading control in multiple publications in the field [see e.g. Nozue et al. *Nature* 448, 358-U311 (2007) and Kumar et al. *Nature* 484, 242-245 (2012)]. Considering the wide acceptance of this method we consider this parameter sufficient for internal protein loading control.

b. To show FRS genes are the clock-regulated, the authors entrained Arabidopsis protoplasts in 12L/12D and transferred continuous red light, which is unusual condition, and used transient luciferase report system. Any idea why the period is lengthened in the subsequent day in the condition?

Also, are there any reasons not to use continuous white light or dark? Or any specific reasons to use only red light?

I recommend that entrain wild type plants, transferred to either continuous white light or dark and harvest at indicated age and time points. And then run qRT-PCR.

The point of the assay was simply to show that the promoter is under circadian control. A qPCR assay will test mRNA levels but not promoter activity. The assay conditions described have been used in multiple publications and are sufficient and adequate for the demonstration of circadian regulation of promoter activity [see e.g. Kim & Somers. *Plant Physiol.* 154, 611-621 (2010); Kim et al. *Proc. Natl. Acad. Sci. U.S.A.* 108, 16843-16848 (2011); Wang et al. *Proc. Natl. Acad. Sci. U. S. A.* 110, 761-766 (2013)]. A different light quality or darkness would only change the period of the oscillation, not the fact of oscillation itself. Under free-running conditions (constant light or dark; constant temperature) circadian period is typically longer or shorter than 24 hours, hence the subsequent peaks under constant light will "drift" later relative to the subjective dawn, if the period is longer than 24 hrs.

c. In the manuscript, Arabidopsis protoplasts and Arabidopsis cell culture were used. I wonder if those are appropriate materials for the experiments.

It stands beyond doubt that it would be hard to defend a theory solely based on assays in cell cultures or protoplasts. However, all of our assays in these systems are routinely used also in this field, and, most importantly, the data obtained have been corroborated through various complementary experiments *in planta*. Hence, accordingly, we are confident about the relevance and the appropriateness of the results obtained in these systems.

Reviewer #3:

The authors have characterized FRS7 and FRS12, which are circadian and photoperiodic regulated proteins, in this study. These proteins highly accumulate in short-day conditions and have a role in transcription factor. They have demonstrated that these proteins repress hypocotyl growth and regulate flowering time in a photoperiodic-dependent manner using mutant lines and overexpression lines. They have also found some interacting proteins with FRS12 by IP-MS analysis using TAP-tag purification method expressed in cultured Arabidopsis cells. It was confirmed that these candidate proteins were localized with FRS12 in nucleus by BiFC assay. Then, they have found that FRS12 bind to cis-elements on promoter region of some downregulated genes in FRS12 overexpression lines. Please see following comments for improve manuscript.

1. They have showed only 4 identified proteins by IP-MS analysis in Figure 4a and Supplementary Data 1. I would like to make sure that other proteins were not identified or not. Because more proteins including false-positive proteins are usually identified in proteomic analysis using LTQ-Orbitrap Velos. If they choose only these 4 proteins, please show the criteria.

Indeed many of the identified proteins after TAP are nonspecific interactors. They are high abundant and promiscuous proteins present in a lot of TAP eluates. These proteins are filtered out based on frequency of occurrence of the co-purified proteins in a large dataset containing 543 TAP experiments using 115 different baits, as published by Van Leene et al, 2015. This is now described in short in the methods section (see page 31 lines 536-544). The full criteria can be found in Van Leene et al, 2015 [Nat. Protoc. 10, 169-187] to which we refer in the methods section. Furthermore, a new table with the specific peptides for the four identified FRS12-interacting proteins has been added (Supplementary Table 2; see also the comment below).

2. I recommend that they should make a new Table and show if unique peptides were assigned to both FRS7 and FRS12. Otherwise, it cannot exclude that only conserved peptides between FRS7 and FRS12 were identified. They should also show unique peptides were assigned to AHL9 and AHL14.

The requested data was already part of the .xlsx file 'Supplementary Data 1'. However, for clarity, as suggested by the reviewer, a new table (Supplementary Table 2) was created showing the list of all unique peptide sequences assigned to FRS12 and its interactors (see also our reply to the comment above).

3. P7. L12-14: I could not understand the meaning of "Saturnalia Complex (SC)". Does this name make sense based on their function? If I correctly understand of this meaning, it is not suit to their function.

We agree with the reviewer. Based on the function of FRS7 and FRS12, which is further supported by the data obtained through the additional experiments demanded by all

reviewers, the name “Saturnalia Complex” did not suit their function. It has therefore been replaced by the “FRS7-FRS12 complex”.

4. Concerning RNA-seq results, they should describe up-regulated genes bit more. I feel that they just describe down-regulated genes because these genes are convenience to explain their model.

A paragraph summarizing the main observations of GO analysis of the upregulated genes has been included in the results section (see page 8 lines 188-194).

5. P25 (TAP method): They describe the TAP methods very simply in this manuscript. They just cite ref. 26 as method. I would like to ask them if their methods concerning IP-MS are exactly same with ref. 26. How did they prepare Arabidopsis cell cultures with FRS12-TAP? Have they performed in-gel digestion? If so, how many gels were excised? Have they used exactly same HPLC, buffer, column, method and so on? If they modify concerning some steps, they should describe details. Otherwise, readers cannot reproduce their experiments.

Yes, we have followed exactly the same procedure as that described by Van Leene et al. 2015 since the TAP technology exists as an in-house service platform with precise standard operating procedures, which therefore allows comparing experiments with different baits carried out at different times.

6. Please explain briefly concerning min35S:flUC method. I could not understand correctly this method.

The *CaMV35S* minimal promoter region, consisting of the region -46 to -1 of the 35S promoter, does not drive the expression of a gene by itself but acts as an enhancer of regulatory activation elements in the promoters of interest. As such it has often been used to enhance basal expression levels of otherwise weak gene promoters in gene discovery projects and/or promoter studies. In this context, we fused the minimal *CaMV35S* promoter region to the 3' end of the *PIF4* promoter in order to enhance endogenous activation of this promoter. This has now been explained better in the methods section (page 33-34 lines 591-594).

7. P6, Line 8, Co-0 should be Col-0.

The text has been corrected accordingly.

Reviewer #4:

The authors proposed that FRS7 and FRS12 proteins are involved in seasonal growth and flowering time. To conclude this, they performed quite broad experiments, most of which were with cutting edge methods (gene expression analysis, ChIPseq analysis, gene reporter assay, and phenotypes observation). This is first time to report that these FRS genes are implicated in seasonal responses in plants. The paper is well written and easy to follow, and may have impact on related research fields.

However, I have some concerns about data in the manuscript.

1. Although authors described that frs7 frs12 double mutants and over-expression lines of FRSs showed opposite phenotypes, but they used only one allele of frs7 and frs12, and any complementation tests were not performed. Phenotypes of mutants are likely, but other genetic loci are possibly responsible for the phenotypes.

Unfortunately, at the beginning of this study, we could not obtain additional alleles as they were either not available (for *FRS7*) or did not show suppressed gene expression (for *FRS12*). Nonetheless, considering the observed additive phenotypes of the double *frs7-1;frs12-1* line as compared to the single mutants, we believe the causative effect of the *frs7-1* and *frs12-1* mutations is likely. Furthermore, all single *frs7-1*, *frs12-1* and double *frs7-1;frs12-1* lines were backcrossed in the wildtype Col-0 background for four generations. Although this procedure may not completely rule out the possibility of other causative genetic loci, it will strongly reduce it.

We are currently in the process of constructing additional *frs7;frs12* KO lines using the CRISPR-CAS9 technology. As these lines are still segregating, we could not include analysis with such lines in due time. However, in a preliminary experiment with the segregating material, these lines show already overly elongated hypocotyl phenotypes under SD growth (see the figure included below in this response letter, provided for the information of the reviewer). These preliminary data confirm the previously observed phenotypes of the *frs7-1;frs12-1* double mutant. Together with the observed opposed phenotypes of over-expressing lines and the extensive molecular data presented in the manuscript, we therefore believe strongly that other genetic loci are not responsible for the phenotypes.

Figure 1. FRS7 and FRS12 repress hypocotyl growth. Hypocotyl length measurements of Arabidopsis Col-0 wt seedlings compared to gain- and loss-of-function lines of *FRS7* and *FRS12* grown for 10 days in SD. Values represent the average of 30 biological replicates \pm SE (*** P <0.001, ** P <0.01, * P <0.05 t-test).

2. They described that expression of *FRS7* and *FRS12* are under the clock control, from the luminescence reporter assay using mesophyll protoplasts. However, public database (DIURNAL, <http://diurnal.mocklerlab.org/>) show constant mRNA amounts of these genes under diurnal and circadian conditions. In addition, amplitude of luminescence rhythms are relatively lower, compared with those driven by canonical clock genes promoters. *FRS12* promoter activity is mainly in leaf vasculature, not in mesophyll (Figure 3a). I feel that more careful description are required for expression of these FRSs.

Based on additional data obtained through the experiments suggested by reviewer#1 (see comment#1) and reviewer#2 (see comment#1), we have described the diurnal transcription patterns of *FRS7* and *FRS12* more accurate now (see the revised Figure 1). The new expression analysis was carried out in whole seedling and confirms our luminescence reporter assays.

We agree with the reviewer that compared to a canonical clock component the diurnal expression amplitudes of *FRS7* or *FRS12* might appear low. However, as discussed in the manuscript neither *FRS7* nor *FRS12* are clock components, but rather outputs of the clock, explaining their comparatively reduced amplitudes.

3. They performed ChIP-seq (TChAp-seq) using cultured cell, but it is not clear whether *FRS12*-HBH fusion protein expressed in these cells are biologically active (native *FRS12* is expressed in vasculature). This may call claim the result of TChAp-seq (Do binding sites are reflected of biological function of *FRS12*?). Information about reads (or depth) of mapping data (Figure 5) are not shown.

We have added additional information about total sequencing coverage and read mapping in Supplementary Table 6.

As indicated in our response to comment#6c of reviewer#2, we agree that one must be careful when drawing conclusions from single experiments in a particular model system. However, also here, several lines of evidence support the biological relevance of our TChAP-Seq results:

- TChAP-Seq comparative analyses presented in Supplementary Figure 6 clearly point out functions of FRS12 in flowering development and growth, which is consistent with our observed phenotypes and RNA-Seq analyses.
- The binding site FRS12 box 1 (CACACA) is similar to that of its homolog FHY3 (CACGCGC) [see Ouyang et al. *Plant Cell* 23, 2514-2535 (2011) and Li et al. *Proc. Natl. Acad. Sci. U. S. A.* 113, 9375-9380 (2016)].
- Both FRS7 and FRS12 bind to sites highlighted in the TChAP-Seq analysis. This was confirmed *in planta* by ChIP-qPCR analysis using lines expressing each of these transcription factors under regulation of their own promoters (Figure 6d and Supplementary Figure 8).
- FRS7 and FRS12 are capable of repressing target promoters identified in the TChAP-Seq analysis, as shown in Figure 6e-g.
- The binding of FRS7 to these target regions correlates with the diurnal accumulation of the protein (Figure 1c) as shown in Supplementary Figure 8.

4. Related to TChAp-seq. The manuscript contains comparison data among previous ChIPseq studies and FRS12 ChIPseq, but lacks the accurate knowledge for gene regulatory network under the clock (ChIPseq study of CCA1, TOC1, and PRRs; <http://www.ncbi.nlm.nih.gov/pubmed/?term=chip%2C+circadian%2C+arabidopsis>). PIF4 gene is bound by TOC1 and PRRs, and GI gene is bound by CCA1. Comparison of previous studies and this manuscript may provide further insight for gene regulatory network for photoperiodic flowering time and growth control.

As suggested by the reviewer, we have included a gene regulatory network describing the co-bound target genes by FRS12, the clock and PIF4 (new Supplementary Figure 7).

5. They suggest that regulation of GI expression by SC is involved in photoperiodic flowering pathway. In this scenario, expression of FKF1, a critical partner of GI for flowering induction, is also important. However, FKF1 expression in *frs* mutants are not shown.

As requested by the reviewer, we have now investigated FKF1 expression in the double *frs7-1;frs12-1* mutant, and the *Pro35S:FRS7-HA-1* and *Pro35S:FRS12-HA-1* overexpressing lines grown both under SD and LD. The results have been included in Supplementary Figure 9.

6. Expression of PIF genes for light and clock dependent growth is mainly in epidermis (reviewed in, <http://www.ncbi.nlm.nih.gov/pubmed/26723003>), and this is important for growth. However, FRS expression are in vascular, which may not control PIF expression directly in epidermis.

To the best of our knowledge the reference cited by the reviewer does not mention *PIF4* or its spatial expression, nor are we aware of papers that indicate that *PIF4* has epidermis specific expression. Conversely however, we would like to refer the reviewer to the manuscript by Kumar et al. 2012 [Nature 484, 242-245], which demonstrates that *PIF4* is expressed in vascular tissue.

7. They suggest that *FRS* proteins function in dark period, but expression changes of *GI* and *PIF4* (*FRS*-targets) are in daytime (Figure 6). How do they explain the difference of accumulation time of *FRS* and effecting time on *FRS*-targets.

We apologise for the confusion created in the previous version of the manuscript. Previous and new data indicate that *FRS7* and *FRS12* proteins accumulate during daytime, peak at dusk and decrease at night time (Figure 1, see also our replies to comment#1 from reviewer#1 and comment#1 from reviewer#2). This pattern of diurnal expression correlates exactly with the binding activity of *FRS7* (Supplementary Figure 8). Finally, these data also correlate with the altered expression profiles of the *PIF4* and *GI* target genes in the *frs7-1;frs12-1* mutant line (Figure 7a). Based on all of these results we have adapted our model of the functioning of *FRS7* and *FRS12* proteins in the revised manuscript.

8. Minor comments.

a. Figure 4e, fold change is ">" 2.

The text has been corrected accordingly.

b. page9, line10, *GI* downstream may be "*API*" downstream.

The text has been corrected accordingly.

c. *PRR7* expression is downregulated by *FRS12-GR* (Figure 4), but it is not changed in *frs7* *frs12* (Figure 6). What do the data tell us?

This result may indicate that *PRR7* is indirectly repressed by *FRS12*. In agreement with this, *PRR7* was not found to be bound by *FRS12* in our TChAP-Seq experiment. Given the complexity and multitude of the regulatory cues that modulate the expression of genes encoding circadian clock components, it is however difficult to 'predict' the behaviour of such an indirect gene in backgrounds with altered expression of only a single regulator.

Reviewers' comments:

Reviewer #1 (Remarks to the Author):

I welcome the response of the authors to my original comments. I think the revised manuscript is greatly improved. I address the replies to each of my three previous points below. In each case, my original point is addressed but, also, in each case, I have raised a more minor supplementary issue that I would like the authors to address.

I am happy that the data as presented are now more internally consistent with respect to the timing of peaks of transcription, transcript levels and protein levels. The new data for FRS7 and FRS12 transcript levels in diurnal cycles showing higher transcript levels in SD than LD can now explain the higher protein levels in SD without need to invoke light specific degradation. The data showing light dependent degradation of FRS7 and FRS12 proteins in etiolated seedlings were removed. It is clear that any light-dependent degradation of the proteins does not prevent the daytime accumulation of the protein and is, therefore, not central for this part of this story. It is possible, however, that the light dependent degradation may be an additional feature slowing the accumulation delaying the protein peak until dusk compared to the transcript peak seen at dawn. (Such a delay in protein accumulation as a result of counteracting degradation is seen in the case of PER protein accumulation in the animal clocks). As I would not favour simply ignoring the data and, in fact, it could be relevant, I would propose the data is reinstated in supplementary data as a possible explanation for the delayed protein peak. The authors state in their rebuttal that the data were removed because the conditions were artificial. Although, I agree, the growth conditions used were unlike those in the other assays in this manuscript, the transfer of etiolated seedlings into light is not entirely artificial and light-dependent degradation was clear so I feel it could pertain to the pattern of delayed protein accumulation.

I am pleased to see that the leaf growth assay in short days has also demonstrated that the mutant line shows larger leaf size/rosette area under these conditions too. The leaf area assay suggests that the phenotype is less dramatic in short days which would seem contradictory to all other data in the manuscript which shows more dramatic effects in short days. The authors acknowledge this in their rebuttal and I feel that they offer a logical explanation for this. However, I think that more discussion along these same lines is required in the manuscript text too in order to address the apparent contradiction. In particular, more emphasis could be made of the fact that the final appearance of the mutant rosette in short days is, in fact, more dramatic than in long days.

The expression assays for downstream genes are greatly improved. The data are now consistent between assays. Here, I would also feel that more emphasis could be made in the text of the enhancement of the mutant phenotype in short days. In all assays, PIF4 and GI expression are more dramatically affected in SD. However, in the short day assays for both PIF4 and GI, while the data for the double mutant show higher expression, the data for the single mutants show lower expression for both PIF4 and GI at dusk in SD suggesting the opposite phenotype in single versus double mutants. How do the authors reconcile this with the fact that the mutants tend to have similar physiological phenotypes to the double

mutant?

Overall, I stand by my original comments that this is a convincing and impressive body of work, revealing a completely new mechanism of regulation in plants which is clearly an important part of appropriate adaptation to photoperiod.

To reiterate my previous comments: The work has important implications across a wide range of plant science research in addressing both growth and flowering and these photoperiodic responses are an environmental adaptation that has important agricultural implications. Both the quantity and quality of the data is impressive. The methodology is entirely valid and sound scientific method is applied to reveal a complete pathway. The findings are internally consistent and all appropriate controls are included. The presentation of the work is also of very high quality. Abstract, introduction and conclusion are all appropriate and the resultant manuscript is clear, easy to follow and accessible to a wide audience.

Reviewer #2 (Remarks to the Author):

Major points

1. Based on Rubisco staining, protein amount loaded in each time point is not even. The authors show that FRS7 protein is the most abundant at ZT 8 in SDs in fig 1c but the protein amount at ZT 0, 4, 8 in short days in fig 1e looks similar, meaning that the abundance of FRS7 protein is not cycling in this conditions. It seems that Rubisco staining doesn't work in this case.
2. The authors claim that the FRS7-12 complex is active during the light period. However, based on gene expression profiles of many genes in fig 7a and fig S9, night time expression seems to be affected by mutations and overexpression of FRS7 and FRS12 as well. Also, as shown in fig S8, FRS7 protein binds to the PIF4 promoter region at ZT 20 in long days.
3. The authors didn't provide photoperiod-dependent abundance changes in the FRS7-12 complex but conclude that the repression activity of FRS7-12 complex is high in short days compared to in long days. However, expression profiles of many genes analyzed in fig 1, fig S9 and fig S10 are still changed by *frs7*, *frs12*, *frs7/12* mutant lines and FRS overexpression lines under long day conditions. Also, Based on the ChIP assay in fig S8, the binding of FRS7 to the promoters of target genes is more strong in LD compared with in SD.
4. The effect of double mutations is much bigger in flowering and hypocotyl length changes compared to each single mutation. However, in fig S10, 1) GI expression is much reduced in a single mutant while increased in the double mutant in short days. 2) PIF4 expression in the *frs7* or *frs12* mutant is higher than in the double mutant.
5. Unfortunately, we cannot quantify the effect of gene expression changes on a target gene

expression. However, based on Figure 7a and Fig S8, CO mRNA expression in the double mutant and ox lines under both LD and SD doesn't correlated with GI expression changes, which doesn't support the authors response "Regarding GI, previous data show that overexpressing this gene in Arabidopsis causes dramatic acceleration in flowering time, especially under short days [see Mizoguchi et al. Plant Cell 17, 2255-2270 (2005)]. Furthermore, these authors showed that GI overexpression causes upregulation of CO without affecting FKF1 expression. Considering these findings, we believe our interpretation about GI is still valid."

Minor point

1. Related to protein profiles, it should be more careful to conclude that FRS7 and 12 genes are clock-regulated. Need more biological replications.
2. The authors are saying that "FRS12 recruits FRS7s in all conditions" but how did the authors conclude it?
3. In fig S8, 'FRS7 and FRS12 bind and repress' should be changed as 'FRS7 binds and represses'

Reviewer #3 (Remarks to the Author):

The authors have revised according to my comments in the latest version. I have no comments anymore.

Reviewer #4 (Remarks to the Author):

The revised version is well written and easy to follow. The data presented here well support authors idea. This study has impact on related research fields.

I have few concerns.

Figure1 in response letter shows that frs12_Crispr does not cause further hypocotyl elongation in frs7-1 mutation. The frs12-1 did not show elongated hypocotyls in same figure.

Dual peaks of FKF1 mRNA expression in wild type under short-day conditions in supplemental figure 9 is unlikely. FKF1 mRNA has single peak in the evening under short-day (many papers reported that, and for example, PMID: 14628054). This expression pattern is crucial for day-length determination.

Authors write 'direct' interaction was observed if BiFC was positive. However, other proteins

in cell might be involved in generation of BiFC signal. In vitro experiment without any other proteins can prove 'direct' interaction.

Response to Reviewer #1:

I welcome the response of the authors to my original comments. I think the revised manuscript is greatly improved. I address the replies to each of my three previous points below. In each case, my original point is addressed but, also, in each case, I have raised a more minor supplementary issue that I would like the authors to address.

1/ I am happy that the data as presented are now more internally consistent with respect to the timing of peaks of transcription, transcript levels and protein levels. The new data for FRS7 and FRS12 transcript levels in diurnal cycles showing higher transcript levels in SD than LD can now explain the higher protein levels in SD without need to invoke light specific degradation. The data showing light dependent degradation of FRS7 and FRS12 proteins in etiolated seedlings were removed. It is clear that any light-dependent degradation of the proteins does not prevent the daytime accumulation of the protein and is, therefore, not central for this part of this story. It is possible, however, that the light dependent degradation may be an additional feature slowing the accumulation delaying the protein peak until dusk compared to the transcript peak seen at dawn. (Such a delay in protein accumulation as a result of counteracting degradation is seen in the case of PER protein accumulation in the animal clocks). As I would not favour simply ignoring the data and, in fact, it could be relevant, I would propose the data is reinstated in supplementary data as a possible explanation for the delayed protein peak. The authors state in their rebuttal that the data were removed because the conditions were artificial. Although, I agree, the growth conditions used were unlike those in the other assays in this manuscript, the transfer of etiolated seedlings into light is not entirely artificial and light-dependent degradation was clear so I feel it could pertain to the pattern of delayed protein accumulation.

Because we wanted to verify the results originally included in the first submission, we repeated the light-dependent degradation experiment, with both FRS7 and FRS12 overexpressing lines and additional relevant controls, and with Actin immunoblotting for loading control instead of Rubisco for instance (see also comment#1 of reviewer#2). As such, we could not reproduce the outcome of the experiment originally included in the first submission. Hence, we believe that at this stage this type of experiment would not add relevant information to our story, nor provide an explanation for the higher protein accumulation under SD conditions. A more detailed study of possible light-mediated regulation of FRS7/12 stability may be warranted, and could be scope of future follow-up research.

2/ I am pleased to see that the leaf growth assay in short days has also demonstrated that the mutant line shows larger leaf size/rosette area under these conditions too. The leaf area assay suggests that the phenotype is less dramatic in short days which would seem contradictory to all other data in the manuscript which shows more dramatic effects in short days. The authors acknowledge this in their rebuttal and I feel that they offer a logical explanation for this. However, I think that more discussion along these same lines is required in the manuscript text too in order to address the apparent contradiction. In particular, more

emphasis could be made of the fact that the final appearance of the mutant rosette in short days is, in fact, more dramatic than in long days.

We have adapted the manuscript text accordingly.

3/ The expression assays for downstream genes are greatly improved. The data are now consistent between assays. Here, I would also feel that more emphasis could be made in the text of the enhancement of the mutant phenotype in short days. In all assays, PIF4 and GI expression are more dramatically affected in SD. However, in the short day assays for both PIF4 and GI, while the data for the double mutant show higher expression, the data for the single mutants show lower expression for both PIF4 and GI at dusk in SD suggesting the opposite phenotype in single versus double mutants. How do the authors reconcile this with the fact that the mutants tend to have similar physiological phenotypes to the double mutant?

We agree that values might appear lower for *GI* and *PIF4* in the single mutants as compared to wt plants. However, we would like to note that these transcript profiling results are quite variable, as illustrated by the new set of Supplementary Figures 10 to 12, showing the data of several independent experiments for the double mutant. Although the trend is similar across independent experiments (see in particular the new Supplementary Figure 11), the differences are statistically not always significantly different, therefore we do not want to emphasize the observed differences too much. Accordingly we have also moderated our discussion on the difference in *PIF4* and *GI* transcripts between the double mutant and the wt. Likewise, we agree that single mutant phenotypes (especially *frs7-1*; the single *frs12-1* mutant never shows any phenotype in our assays) don't seem to be explained at the molecular level by *PIF4* or *GI* de-regulation. However, as specified in the revised manuscript, the FRS7-FRS12 complex may also regulate a large portion of *PIF4* and *GI* downstream genes. It's therefore likely that phenotypes in single and double mutants cannot simply be explained only by the de-regulation of these two known major regulators but also involve the de-regulation of additional downstream genes, or even yet unknown regulators, that were not assessed in this study.

Overall, I stand by my original comments that this is a convincing and impressive body of work, revealing a completely new mechanism of regulation in plants which is clearly an important part of appropriate adaptation to photoperiod.

To reiterate my previous comments: The work has important implications across a wide range of plant science research in addressing both growth and flowering and these photoperiodic responses are an environmental adaptation that has important agricultural implications. Both the quantity and quality of the data is impressive. The methodology is entirely valid and sound scientific method is applied to reveal a complete pathway. The findings are internally consistent and all appropriate controls are included. The presentation of the work is also of very high quality. Abstract, introduction and conclusion are all appropriate and the resultant manuscript is clear, easy to follow and accessible to a wide audience.

We thank the reviewer for these nice words.

Response to Reviewer #2:

Major points

1. *Based on Rubisco staining, protein amount loaded in each time point is not even. The authors show that FRS7 protein is the most abundant at ZT 8 in SDs in fig 1c but the protein amount at ZT 0, 4, 8 in short days in fig 1e looks similar, meaning that the abundance of FRS7 protein is not cycling in this conditions. It seems that Rubisco staining doesn't work in this case.*

We agree with the reviewer and thank him for this observation. We have therefore repeated the experiment to determine diurnal and photoperiodic protein abundance for both *ProFRS7:FRS7-HA* and *ProFRS12:FRS12-HA* lines and using immunoblotting for Actin as the loading control. In these new experiments we could not observe pronounced diurnal changes at the protein level at the different time points tested, either for FRS7 or FRS12 (Figure 1 c-d), indicating that both proteins show rather stable levels over the day/night cycle. In contrast, the higher accumulation when seedlings were grown under SD conditions in comparison to LD conditions remained clear at all time points (Figure 1 e). Based on these new and more accurate results we therefore postulate in the revised manuscript that photoperiodic rather than diurnal control of protein accumulation is the most pronounced regulatory checkpoint. Furthermore, we have additionally included a picture of a wild type seedling (Col-0) in this immunoblot analysis, to establish the utility of the two *ProFRS:FRS-HA* lines for the reader, not only for this immunoblot analysis, but also for the ChIP experiments (see Figure 6 and Supplementary Figure 9) and thereby to support some other revisions of our manuscript (see also comment #2-3 of this reviewer). As shown in Fig 1c-d, it becomes clear that relative accumulation of tagged FRS7 in the *ProFRS7:FRS7-HA* line was much lower than that of tagged FRS12 in the *ProFRS12:FRS12-HA* line. For this reason and in accordance with most of our molecular data presented (TAP, RNA-Seq and TChAP-Seq) we decided to use the *ProFRS12:FRS12-HA* to address comments #2 and #3 of the reviewer.

2. *The authors claim that the FRS7-12 complex is active during the light period. However, based on gene expression profiles of many genes in fig 7a and fig S9, night time expression seems to be affected by mutations and overexpression of FRS7 and FRS12 as well. Also, as shown in fig S8, FRS7 protein binds to the PIF4 promoter region at ZT 20 in long days.*

We thank the reviewer for pointing to these discrepancies in our data. The text has been corrected accordingly and a new ChIP-qPCR experiment has been performed. As

indicated in the response to comment#1 of this reviewer, we noticed in our immunoblot analysis that FRS7-HA expression in our *ProFRS7:FRS7-HA* lines was very weak and FRS7-HA detection in total protein extracts therefore often challenging (Figure 1c). Therefore we decided to remove all the previous ChIP data with this line from the revised manuscript and repeated all of the demanded ChIP experiments with the more robust *ProFRS12:FRS12-HA* line (Figure 1d) in a single experiment to allow both diurnal and photoperiodic comparisons. These new ChIP data are shown in the new Supplementary Figure 9. Enriched binding of FRS12 to *ProPIF4* and *ProGI* was particularly observed when *ProFRS12:FRS12-HA* seedlings were grown in SD conditions, but was independent of the time of the day. These data appear more consistent with the expression data pointed out by the reviewer. Hence, based on our additional, improved, immunoblot and ChIP analysis, we postulate in the revised manuscript that the FRS7-FRS12 complex is present throughout the diurnal cycle but preferentially accumulates under SD-photoperiods, in which it appears more active as well.

3. The authors didn't provide photoperiod-dependent abundance changes in the FRS7-12 complex but conclude that the repression activity of FRS7-12 complex is high in short days compared to in long days. However, expression profiles of many genes analyzed in fig 1, fig S9 and fig S10 are still changed by frs7, frs12, frs7/12 mutant lines and FRS overexpression lines under long day conditions. Also, based on the ChIP assay in fig S8, the binding of FRS7 to the promoters of target genes is more strong in LD compared with in SD.

We apologise for the confusion. The previous ChIP assay did not allow comparison between SD and LD as these were data from independent analysis. However, as indicated above, the new protein accumulation and ChIP data that now allow such comparison, demonstrate that the FRS7-FRS12 complex preferentially accumulates under SD-photoperiods, in which it appears more active as well, but nonetheless also accumulates in LD photoperiods where it is likely also active, given the phenotypes in LD conditions. However, in the latter conditions, it may not necessarily be acting on the same set of targets or with the same partners.

4. The effect of double mutations is much bigger in flowering and hypocotyl length changes compared to each single mutation. However, in fig S10, 1) GI expression is much reduced in a single mutant while increased in the double mutant in short days. 2) PIF4 expression in the frs7 or frs12 mutant is higher than in the double mutant.

These concerns have also been raised by reviewer #1, please see our reply to comment#3 of reviewer#1 for more details.

5. Unfortunately, we cannot quantify the effect of gene expression changes on a target gene expression. However, based on Figure 7a and Fig S8, CO mRNA expression in the double mutant and ox lines under both LD and SD doesn't correlated with GI expression changes, which doesn't support the authors response "Regarding GI, previous data show that

overexpressing this gene in Arabidopsis causes dramatic acceleration in flowering time, especially under short days [see Mizoguchi et al. Plant Cell 17, 2255-2270 (2005)]. Furthermore, these authors showed that GI overexpression causes upregulation of CO without affecting FKF1 expression. Considering these findings, we believe our interpretation about GI is still valid.”

We agree that CO mRNA expression does not correlate with GI expression changes in the double mutant line, suggesting that the observed flowering phenotypes are possibly regulated by additional flowering time regulatory pathways such as those modulated by PIF4 (as suggested in the revised discussion). Nonetheless, our data clearly demonstrate that FRS7/FRS12 are able to repress and bind GI, hence they support our hypothesis that the FRS7-FRS12 complex can, at least partially, regulate the GI-CO pathway by binding and repressing GI under SD photoperiods.

Minor point

1. Related to protein profiles, it should be more careful to conclude that FRS7 and 12 genes are clock-regulated. Need more biological replications.

We agree with the reviewer that the protein profiles do not insinuate clock-regulation of FRS7-FRS12, therefore we have moderated the manuscript text accordingly. However, we do believe that our circadian bioluminescence assays (Fig. 1a) clearly support that the promoters of FRS7 and FRS12 are under regulation of the circadian clock, given the robust rhythmicity expression pattern of the fLUC reporter gene. As it was already mentioned in our previous response letter to reviewers, the assay conditions described have been used in multiple publications and are sufficient and adequate for the demonstration of circadian regulation of promoter activity [see e.g. Kim & Somers. Plant Physiol. 154, 611-621 (2010); Kim et al. Proc. Natl. Acad. Sci. U.S.A. 108, 16843-16848 (2011); Wang et al. Proc. Natl. Acad. Sci. U. S. A. 110, 761-766 (2013)]. This experiment was performed in 6 biological replicates (now specified in the materials and methods section). Hence, clock-regulation of FRS7 and FRS12 does seem to exist, but may not always be tractable at all levels.

2. The authors are saying that “FRS12 recruits FRS7s in all conditions” but how did the authors conclude it?

We apologise for the confusion, we referred to all tested conditions. The text has been corrected.

3. In fig S8, ‘FRS7 and FRS12 bind and repress’ should be changed as ‘FRS7 binds and represses’

This figure has been now replaced with new data (see also major comments #2-3 of the reviewer) and it is presented as Supplementary Fig. 9.

Response to Reviewer #3:

*The authors have revised according to my comments in the latest version.
I have no comments anymore.*

Response to Reviewer #4:

*The revised version is well written and easy to follow. The data presented here well support authors idea. This study has impact on related research fields.
I have few concerns.*

1/ Figure1 in response letter shows that frs12_Crispr does not cause further hypocotyl elongation in frs7-1 mutation. The frs12-1 did not show elongated hypocotyls in same figure.

As indicated in our previous response letter, that preliminary hypocotyl elongation experiment was conducted with CRISPR lines that were still segregating and limited biological repeats. To avoid variability, we have now repeated this experiment with a larger amount of biological replicates as well as with CRISPR plants from the next generation (see the revised Figure 2a-b and the new Supplementary Figure 2). First, the obtained data indeed indicate no significant differences in elongation in the *frs12-1* mutant compared to Col-0 wt plants. In contrast, the single *frs7-1* mutant presents significantly elongated hypocotyls, in SD only. This phenotype is notably accentuated in the double *frs7-1;frs12-1* line, suggesting a cooperative function of FRS7 and FRS12. Second, similar to the *frs7-1;frs12-1* double mutant, the independent double mutant line generated through CRISPR, also presented increased hypocotyl elongation specifically under SD-growth conditions. Together these results confirm the coordinated functions of FRS7 and FRS12 to regulate hypocotyl growth in a photoperiodic dependent manner.

2/ Dual peaks of FKF1 mRNA expression in wild type under short-day conditions in supplemental figure 9 is unlikely. FKF1 mRNA has single peak in the evening under short-day (many papers reported that, and for example, PMID: 14628054). This expression pattern is crucial for day-length determination.

We thank the reviewer for this observation. The complete diurnal oscillation expression analysis was repeated for *FKF1* and *PIL1* and the inappropriate peaks were no longer present. We sincerely apologize for the erroneous presentation in our previous figure. We believe it must have been due to a technical error during the performance of the qPCR analysis. A revised Supplementary Figure (now Supplementary Figure 10) with the correct oscillation patterns is now provided. In addition, we also noticed a technical error during the calculation of our qPCR data from Figure 1b (*FRS7* expression). The conclusions of these experiments do not change but a new corrected figure is now provided.

3/ Authors write 'direct' interaction was observed if BiFC was positive. However, other proteins in cell might be involved in generation of BiFC signal. In vitro experiment without any other proteins can prove 'direct' interaction.

We agree with the reviewer. The text has been corrected accordingly.

REVIEWERS' COMMENTS:

Reviewer #1 (Remarks to the Author):

The latest version of "The transcriptional repressor complex FRS7-FRS12 regulates flowering time and growth in Arabidopsis" satisfactorily addresses all of my previous queries. I am happy that the revised manuscript now forms a cohesive body of work. The paper describes a significant new mechanism in the regulation of plant growth and development in response to environmental stimuli and so forms an important contribution to our knowledge.

Two minor grammatical corrections:

Line 275. "... we questioned if FRS12 could attenuate binding to its targets...". "Attenuate" is a transitive verb (requiring an object) but, here, the authors do not refer to FRS12 acting on something else. Perhaps "... we questioned if FRS12 could show altered binding to its targets ...".

Line 323. "Alternatively, given the weak and variable effect on ...". Remove the word "alternatively". This is not an alternative to the previous sentence as is suggested by the use of the word "alternatively" here.

Reviewer #2 (Remarks to the Author):

The authors have revised according to my earlier comments.
I have no further comments.

Reviewer #4 (Remarks to the Author):

The revised manuscript were well improved and also satisfied my comments. I have no concern.

Response to Reviewer #1

Two minor grammatical corrections:

1/ Line 275. "... we questioned if FRS12 could attenuate binding to its targets...". "Attenuate" is a transitive verb (requiring an object) but, here, the authors do not refer to FRS12 acting on something else. Perhaps "... we questioned if FRS12 could show altered binding to its targets ...".

2/ Line 323. "Alternatively, given the weak and variable effect on ...". Remove the word "alternatively". This is not an alternative to the previous sentence as is suggested by the use of the word "alternatively" here.

Both corrections have been implemented.